# SMOS-derived Antarctic thickness of thin sea ice: data description and validation in the Weddell Sea

Lars Kaleschke[1,*], Xiangshan Tian-Kunze[1,*], Stefan Hendricks[1], and Robert Ricker[2]

[1]Alfred Wegener Institute, Helmholtz Centre for Polar and Marine Research, Bremerhaven, Germany
[2]NORCE Norwegian Research Center, Tromsø, Norway
[*]These authors contributed equally to this work.

**Correspondence:** Lars Kaleschke (lars.kaleschke@awi.de)

**Abstract.** Accurate satellite measurements of the thickness of Antarctic sea ice are urgently needed but pose a particular challenge. The Antarctic data presented here were produced using a method to derive the sea ice thickness from 1.4 GHz brightness temperatures previously developed for the Arctic, with only modified auxiliary data. The ability to observe the thickness of thin sea ice using this method is limited to cold conditions, meaning it is only reasonable during the freezing period, typically March to October. The SMOS level 3 sea ice thickness product contains estimates of the sea ice thickness and its uncertainty up to a thickness of about 1 m. The sea ice thickness is provided as daily average on a polar stereographic projection grid with a sample resolution of 12.5 km, while the SMOS brightness temperature data used has a footprint size of about 35-40 km in diameter. Data from SMOS have been available since 2010, and the mission's operation has been extended to continue until at least the end of 2025.

Here we compare two versions of the SMOS Antarctic sea ice thickness product which are based on different level 1 input data (v3.2 based on SMOS L1C v620, and v3.3 based on SMOS L1C 724). A validation is performed to generate a first baseline reference for future improvements of the retrieval algorithm and synergies with other sensors.

Sea ice thickness measurements to validate the SMOS product are particularly rare in Antarctica, especially during the winter season and for the valid range of thicknesses. From the available validation measurements, we selected datasets from the Weddell Sea that have varying degrees of representativeness: Helicopter-based EM Bird (HEM), Surface and Under-Ice Trawl (SUIT), and stationary Upward-Looking Sonars (ULS). While the helicopter can measure hundreds of kilometers, the SUIT's use is limited to distances of a few kilometers and thus only captures a small fraction of an SMOS footprint. Compared to SMOS, the ULS are point measurements and multi-year time series are necessary to enable a statistically representative comparison. Only four of the ULS moorings have a temporal overlap with SMOS in the year 2010.

Based on selected averaged HEM flights and monthly ULS climatologies we find a small mean difference (bias) of less than 10 cm and a root-mean-square deviation of about 20 cm with a correlation coefficient R>0.9 for the valid sea ice thickness range between zero and about one meter. The SMOS sea ice thickness showed an underestimate of about 40 cm with respect to the less representative SUIT validation data in the marginal ice zone. Compared with sea ice thickness outside the valid range we find that SMOS strongly underestimates the real values which underlines the need for combination with other sensors such as altimeters.

In summary, the overall validity of the SMOS sea ice thickness for thin sea-ice up to a thickness of about 1 m has been demonstrated through validation with multiple datasets. To ensure the quality of the SMOS product, an independent regional sea-ice extent index was used for control. We found that the new version v3.3 is slightly improved in terms of completeness, indicating less missing data. However, it is worth noting that the general characteristics of both datasets are very similar, also with the same limitations. Archived data are available on the PANGAEA repository at https://doi.org/10.1594/PANGAEA.934732 (Tian-Kunze and Kaleschke, 2021), and operationally at https://doi.org/10.57780/sm1-5ebe10b (European Space Agency, 2023).

## 1 Introduction

Antarctic sea ice has very different characteristics compared with that in the Arctic (Eicken et al., 1995; Wever et al., 2021). Antarctic sea ice is dominated by first-year ice, with multi-year ice mainly distributed in the western Weddell Sea (Worby et al., 2008). The extent of Antarctic sea ice includes twice as much seasonal sea-ice as in the Arctic, but half as much perennial ice as in the Arctic (Stammerjohn and Maksym, 2017). In contrast to the dramatic sea-ice retreat observed in the Arctic the last two decades (Stroeve et al., 2012; Meier et al., 2014), the sea ice extent in the Antarctic showed little change or even slight increasing trends (Zhang, 2007; Holland et al., 2014; Parkinson and Cavalieri, 2012; Allan et al., 2021). However, the situation has recently changed, with the record-low sea-ice extent anomalies observed in 2022 and 2023 indicating a need for further research (Voosen, 2023; Gómez-Valdivia et al., 2023; Li et al., 2023; Zhou et al., 2023; Purich and Doddridge, 2023; Hobbs et al., 2024).

Although Antarctic sea-ice concentration has been monitored from space since last five decades, information about the thickness has been elusive until the launch of dedicated altimeter missions (Giles et al., 2008; Kurtz et al., 2012; Zwally et al., 2008). sea ice thickness up to 1 m, further referred to as thin ice, can also be estimated from brightness temperatures such as measured by the L-band radiometer on board of ESA's Soil Moisture and Ocean Salinity mission SMOS. However a thorough validation has been carried out so far only for the northern hemisphere (Kaleschke et al., 2012; Tian-Kunze et al., 2014; Huntemann et al., 2014; Maaß et al., 2015; Kaleschke et al., 2016). Due to the broad swath-width of about 1000 km, SMOS provides daily coverage and is thus complementary to altimeter sensors which measure only along a narrow profile (Ricker et al., 2017).

The Scientific Committee on Antarctic Research (SCAR) Antarctic Sea Ice Processes and Climate (ASPeCt) program has collected ship-born observations over two decades (between 1981 and 2005) and provided one of the most informative map about the distribution of sea ice thickness in the Antarctic (Worby et al., 2008). However, due to the harsh conditions in the Antarctic during most time of the year, continuous Antarctic-wide observations are lacking, therefore, it is difficult to assess the seasonal and regional variability based on the sparse estimates from ship observations. Giles et al. (2008) compared the ASPeCt data with seven years ice elevation data from ERS radar altimetry and found general agreement in the climatology, but discrepancies in average sea ice thickness estimates, which could be caused by snow cover and under sampling of ridges in the ASPeCt data.

The influence of snow on the sea-ice, which is often flooded with seawater, leads to high uncertainty for sea ice thickness estimates by radar altimeters such as ESA's CryoSat2 (Laxon et al., 2013; Ricker et al., 2014). Sea-ice freeboard derived from laser altimeters on board of NASA's Ice, Cloud, and land Elevation Satellites (ICESat and ICESat-2) suffers from the large uncertainty in the estimation of snow mass on sea ice (Kern and Spreen, 2015; Kern et al., 2016). Kacimi and Kwok (2020) found that 60-70% of the total freeboard measured by ICESat-2 altimeter consists of snow. sea ice thickness and volume in the Antarctic has been analyzed by (Kurtz et al., 2012) using ICESat laser altimetry data for the period of 2003-2008, assuming that the snow-ice interface is at sea level. No significant trend has been observed. The results were undermined by the large uncertainties in thickness and volume calculation. Xu et al. (2021) investigated the consistency of total freeboard between ICESat and ICESat-2, and used an empirical relation between total freeboard, sea ice thickness, and snow depth to derive a circum-Antarctic sea ice thickness trend over the years 2003 to 2020, with missing data from 2009 to 2018. The CRYO2ICE campaign with optimized orbital overlaps of CryoSat-2 and ICESat-2 will potentially help to derive the snow depth through the difference in penetration but difficulties in laser-radar comparisons will also remain due to the sampling discrepancies (Fons et al., 2021).

There are many different ways to measure the sea ice thickness with different uncertainties applicable on different spatial and temporal scales. The most accurate measurements are the in situ drill holes which allow not only to determine the sea-ice and snow thickness but also the freeboard level. This method is a time-consuming procedure and strictly limited to in situ field campaigns at limited locations, thus not representative for large scale sea ice thickness distributions.

Airborne validation campaigns like NASA's Operation IceBridge or ESA's recent 2022 CryoVEx/DEFIANT Antarctica campaign for the CRYO2ICE missions were always carried out during the summer months for logistical reasons. The SMOS sea ice thickness retrieval method is not applicable in the mild temperatures encountered between November and January and therefore these airborne campaign data cannot be used for validation. Ice-mass balance buoys are usually deployed on relatively thick sea ice exceeding the limit of the SMOS retrieval method and are therefore also not directly usable for the validation (Wever et al., 2021).

Stationary Upward looking sonars (ULS) or Acoustic Doppler Current Profiler (ADCP) allows to determine the ice draft (bottom below water level) over a long period (Behrendt et al., 2015; Belter et al., 2020). Distance measurements from below are more accurate than freeboard measurements by laser/radar altimeter because of the similar density of sea ice and water (Belter et al., 2020). Ice draft measurements can be stationary (moored ULS or ADCP) or along profiles with the Surface and Under-Ice Trawl (SUIT) which has to be drawn by a vessel like Polarstern.

The electromagnetic induction (EM) method allows to measure the total thickness of sea ice and snow with a good accuracy (Haas et al., 2009). The EM method can be applied on a sled, from a ship, or from an airborne platform (fixed-wing aircraft or helicopter). Airborne EM data have the advantage to resolve the sub-grid-scale variability of sea ice thickness within a SMOS footprint while at the same time providing a reasonable representativity. Airborne EM measurements have been compared with SMOS sea ice thickness data in previous studies (Tian-Kunze et al., 2014; Kaleschke et al., 2016). The SMOS sea ice thickness estimate includes a statistical correction to account for the thickness distribution function. Therefore, the mean rather

than the mode of the EM-measured ice thicknesses is compared against the SMOS sea ice thickness. The EM-measured sea ice thickness is the total thickness, i.e. snow+ice, whereas SMOS sea ice thickness does not include the snow layer.

In this paper, we present two different versions of the SMOS-derived pan-Antarctic thin sea ice thickness data from 2010 to 2020 (v3.2) and to 2023 (v3.3). The same retrieval procedure as in the Arctic (Tian-Kunze et al., 2014) was applied here to Antarctica, but with adjusted auxiliary data fields. The retrieved SMOS sea ice thickness is compared with in situ measurements from ULS, SUIT and Helicopter-based EM (HEM) in the Atlantic sector of Antarctic, namely the Weddell Sea. The algorithm version v3.2 is based on SMOS L1C brightness temperature version v620 input data, as a reference for the PANGAEA data repository at https://doi.org/10.1594/PANGAEA.934732. Moreover, we perform quality control using independent passive microwave data. The quality control includes also the current operational SMOS sea ice thickness data, version v3.3, based on SMOS L1C brightness temperature version v724. In addition to quality control, the appendix contains an overview of the different SMOS product versions and further comparisons with independent data (ASPeCt and MODIS), which we do not consider as validation.

## 2 Data

### 2.1 Data used for the SMOS sea ice retrieval

The basis of SMOS sea ice thickness retrieval is the brightness temperature measured by the SMOS payload Microwave Imaging Radiometer using Aperture Synthesis (MIRAS) at L-band. Furthermore, SMOS sea ice thickness retrieval needs two auxiliary data sets: one is atmospheric reanalysis data; the other is sea surface salinity climatology.

### 2.1.1 Gridded L3B SMOS brightness temperature

MIRAS measures the brightness temperatures in full polarization with incidence angles ranging from 0° to 65° (Kerr et al., 2001; Mecklenburg et al., 2016). The hexagon-like, two-dimensional snapshots measure one or two of the Stokes components in the antenna reference frame. Horizontally and vertically polarized brightness temperatures are measured by separate snapshots. SMOS measures brightness temperatures with a spatial resolution of about 35 km at nadir on a daily basis in the polar regions.

Over sea ice the first Stokes parameter, the average of the horizontally and the vertically polarized brightness temperatures, is almost independent of incidence angle in range of 0°-40° and independent of both geometric and Faraday rotations, therefore, robust to instrumental and geophysical errors (Camps et al., 2005). We average the intensities over the incidence angle range of 0°-40° to reduce the uncertainty of single measurement. The daily averaged brightness temperature intensities in the Arctic and in the Antarctic are interpolated with nearest neighbor algorithm and gridded into the National Snow and Ice Data Center (NSIDC) polar stereographic projection with a grid resolution of 12.5 km. The northern and southern boundaries of the polar regions are defined as latitude 50°N and 50°S for the sea ice thickness retrieval. The L3B brightness temperatures are also included in the SMOS L3C sea ice thickness product.

### 2.1.2 Atmospheric reanalysis data

To estimate ice surface temperature, we extract the $2\,\mathrm{m}$ surface air temperature and the $10\,\mathrm{m}$ wind velocity data from the Japanese 55-year Reanalysis (JRA55) (Kobayashi et al., 2015) and interpolate them into the polar stereographic projection with 12.5 km grid resolution. JRA55 reanalysis data provide various physical variables in 1.25° resolution every six hours. For SMOS retrieval we consider three previous days' temperature and wind field data and average them as boundary conditions for a thermodynamic model. We assume in the thermodynamic model an immediate equilibrium at air-ice surface, therefore, the time average resembles a temperature diffusion process towards deeper emitting layers.

### 2.1.3 Sea surface salinity climatology

The Sea Surface Salinity (SSS) climatology in the Antarctic (Figure 1) is based on the monthly model outputs of the German contribution of the Estimating the Circulation and Climate of the Ocean project (GECCO2), a quasi-global simulation using the MIT General Circulation Model (MITgcm) over the years of 1952-2001. The model has a horizontal resolution of 1°x1°, with 23 vertical levels. Various in-situ measurements and satellite data were assimilated using the adjoint method (Köhl and Stammer, 2008). In contrast to the highly variable SSS distribution in the Arctic due to river run-offs, SSS in the Antarctic is relatively constant, slightly varying between 33-35 $\mathrm{gkg}^{-1}$, with a standard deviation less than 1 $\mathrm{gkg}^{-1}$.

## 2.2 Data used for the validation

In this study SMOS sea ice thickness data are compared with HEM, SUIT and ULS measurements conducted in the Weddell Sea.

### 2.2.1 Sea ice thickness measured by HEM

sea ice thickness was measured using the HEM system during one of the rare Polarstern Antarctic winter expeditions ANT-XXIX/6 in 2013. On four days (19, 20, and 21 June, 2013 and 7 July, 2013) measurements have been carried out in areas with first-year ice conditions. The flight track positions are shown in Figure 2. HEM measures the total thickness of snow and ice from the difference of electromagnetically determined ice/water interface and the laser-measured snow surface. The accuracy of HEM measurements is around 10 cm over level ice (Haas et al., 2009).

### 2.2.2 Sea ice thickness derived from ULS

ULS sensors were moored at diverse locations in the Weddell Sea and measured the ice draft over various deployment periods between 1990-2010 (Behrendt et al., 2013; Behrendt et al., 2015). The uncertainty of the ULS ice draft measurements is between $\pm$ 5 cm to $\pm$ 23 cm depending on different correction methods (Behrendt et al., 2013). The opening angle of the acoustic beam results in a measurement footprint of approximately 6–8 m in diameter which is sampled every 2-15 minutes depending on battery and data storage capacities (Behrendt et al., 2013). The total sea ice thickness was derived from ice draft using the empirical relationship

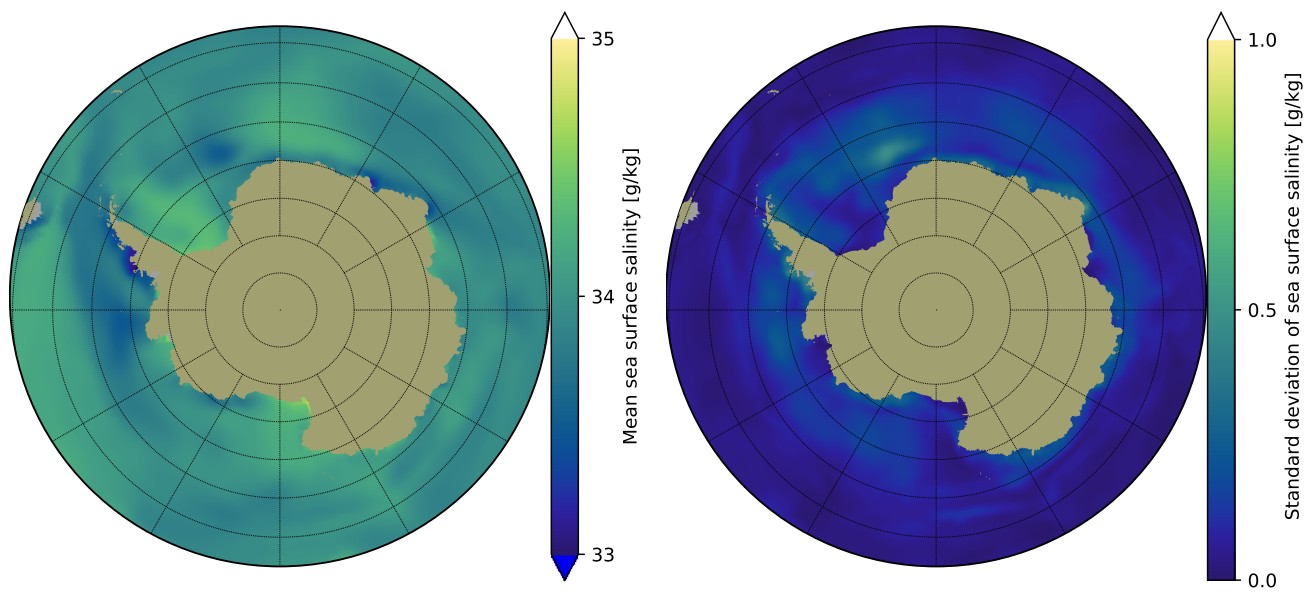

**Figure 1.** Mean (left) and standard deviation (right) of SSS in the Antarctic. The mean and standard deviation are calculated from monthly climatology. Note that the colorbars have different scales.

$$z = 0.028 + 1.012d \tag{1}$$

where $d$ is the sea-ice draft, and $z$ is the total sea ice thickness (Behrendt et al., 2015). Due to the large discrepancy between spatial resolution and sample coverage, it is necessary to compare ULS measurements with SMOS data over longer time peri-
155 ods. A simple arithmetic average is used to calculate the mean values, using all available samples including open water. Three ULS sensors (AWI227, AWI229 and AWI231) have been deployed in areas of predominantly first-year ice and operated for a sufficiently long period of time (Table 1). The ULS AWI230 and AWI210 measured for more than two years in intermediate thick ice conditions and are also used for comparisons. Four ULS (AWI229, AWI206, AWI208, and AWI232) include data for the year 2010 and are therefore also used for a direct comparison to SMOS. The positions of all moorings are shown in Figure
2.

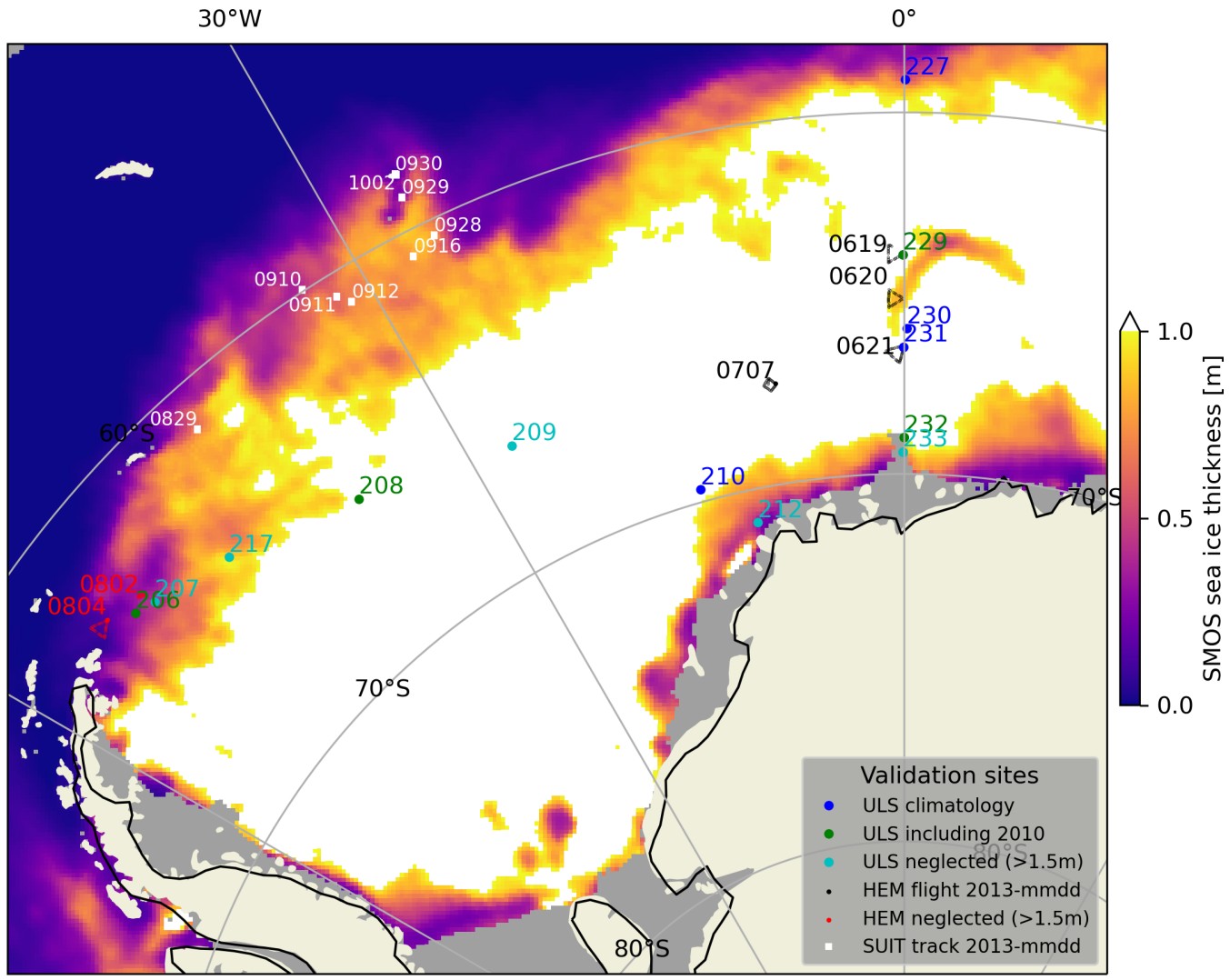

**Figure 2.** Overview of all available validation data used and neglected for the present study. The positions of ULS moorings, SUIT traverses and HEM flights are shown. The background shows the SMOS sea ice thickness averaged for August 2010, a period where some ULS data are still available.

**Table 1.** Available ULS moorings used and neglected in this study. The maximum draft is the maximum of the averaged seasonal cycle calculated from monthly mean values.

| ULS | Latitude [°] | Longitude [°] | Max. draft [m] | # Data [months] | year 2010 |
|-----|--------------|---------------|----------------|-----------------|-----------|
| AWI227 | -59.07 | 0.07 | 0.56 | 47 | |
| AWI229 | -63.97 | -0.05 | 0.75 | 129 | yes |
| AWI231 | -66.51 | -0.03 | 0.88 | 99 | |
| AWI230 | -66.01 | 0.17 | 0.99 | 32 | |
| AWI210 | -69.66 | -15.71 | 1.71 | 25 | |
| AWI206 | -63.48 | -52.10 | 2.79 | 56 | yes |
| AWI208 | -65.61 | -37.41 | 1.72 | 54 | yes |
| AWI232 | -69.00 | -0.00 | 2.01 | 141 | yes |
| neglected | | | | | |
| AWI207 | -63.71 | -50.84 | 2.42 | 87 | |
| AWI209 | -66.62 | -27.12 | 1.20 | 12 | |
| AWI212 | -70.91 | -11.96 | 3.94 | 25 | |
| AWI217 | -64.42 | -45.85 | 2.93 | 25 | |
| AWI233 | -69.39 | -0.07 | 3.07 | 41 | |

### 2.2.3 Sea ice thickness derived from SUIT measurements

SUIT measurements were used to investigate the large-scale variability of physical and biological sea-ice properties during five campaigns in polar oceans (Castellani et al., 2019; Castellani et al., 2020). The Polarstern cruise PS81 was conducted during wintertime (August-October, 2013) in the marginal ice zone of the Weddell Sea and is therefore suited for comparisons with the SMOS sea ice thickness. The sensor array of the SUIT includes an altimeter (Tritech PA500/6-E) incorporated into a Conductivity Temperature Depth (CTD) probe. The sea-ice draft is derived by combining the depth measurements from the CTD pressure sensor with the distance from the sea-ice, and is then corrected with pitch and roll measurements from an Acoustic Doppler Current Profiler (ADCP). The total sea ice thickness was estimated by assuming a fixed density value ($\rho = 917\,\mathrm{kgm}^{-3}$) for sea ice. Figure 2 shows the positions of nine SUIT traverses of different profile lengths ranging from 800 m to 3000 m.

### 3 SMOS sea ice thickness retrieval algorithm

The retrieval algorithm is described in detail in (Tian-Kunze et al., 2014), in the following we summarize the basic principles. The SMOS sea ice thickness is produced using an iterative retrieval algorithm that is based on a thermodynamic sea-ice

model and a radiative transfer model (Figure 3), which takes variations of ice temperature and ice salinity into account. In addition, ice thickness variations within the SMOS spatial resolution are considered through a statistical thickness distribution function derived from high-resolution ice thickness measurements from NASA's Operation IceBridge campaign (Tian-Kunze et al., 2014). The statistical correction takes into account the fact that the mean sea ice thickness is often about twice as large compared to the distribution mode. This retrieval algorithm has been used to generate the operational Arctic SMOS sea ice thickness product at Alfred-Wegener-Institute in the framework of "The SMOS and CryoSat-2 Sea Ice Data Product Processing and Dissemination Service", supported by ESA.

The measured SMOS brightness temperature depends on the ice concentration, the temperatures of the sea and the ice, and their emissivity. The sea ice emissivity mainly depends on the dielectric properties of the snow and ice medium and the roughness of the interfaces. We assume sea ice as a homogeneous medium and neglect scattering which is a reasonable simplification for the wavelength of 21 cm. The modelled sea ice emissivity used for the present retrieval mainly depends on ice thickness $d_{ice}$, ice temperature $T_{ice}$, and ice salinity $S_{ice}$.

The retrieval consists of a simple one-layer radiation model (Menashi et al., 1993) and a thermodynamic model (Maykut, 1986). The radiation model calculates the emissivity of the sea ice layer and the underlying sea water. Brightness temperatures are derived from the emissivity and physical temperatures of sea ice and sea water. Because the emissivity is a function of $T_{ice}$ and $S_{ice}$, the two parameters need to be estimated using auxiliary data. Bulk ice salinity is estimated from sea surface salinity $S_w$ using empirical relationship (Kovacs, 1996). $S_w$ is extracted from Antarctic-wide SSS climatology that is derived from model output (Sect. 2.1.3).

We use bulk ice temperature $T_{ice}$ as the physical sea ice temperature since we apply only one ice layer. The bulk ice temperature is estimated from the thermodynamic model, using auxiliary 2 m air temperature $T_a$ and wind velocity $u$ from atmospheric reanalysis data. Thermal equilibrium is assumed at the surface of the ice layer and the heat fluxes are calculated with a thermodynamic model based on Maykut (1986). Under the assumption of thermal equilibrium, the incoming and outgoing heat fluxes compensate each other (Fig. 3).

The sea ice thickness retrieval scheme is shown in Fig. 4. The processing is performed in three steps:

1. An intermediate brightness temperature dataset L3A is generated from L1C data daily and saved in HDF file format on the local server. These files include all information provided in L1C swath data, in an uniform equal-area grid, the Icosahedral Snyder Equal Area (ISEA 4H9) with 15 km sampling resolution.

2. Gridded L3B brightness temperature data in NSIDC polar-stereographic projection with 12.5 km grid resolution are generated from L3A data for the Northern Hemisphere (NH) and Southern Hemisphere (SH) each.

3. L3C ice thickness is generated from L3B brightness temperatures using a pre-calculated look-up table, with JRA55 reanalysis and sea surface salinity climatology as auxiliary data.

The underestimation bias caused by the assumption of a 100% ice coverage increases with decreasing ice concentration (Tian-Kunze et al., 2014). An estimation of the retrieval bias and uncertainty has been carried out in a previous study (Tian-Kunze et al., 2014) in the Arctic. The attempt to correct the bias with ice concentration data from passive microwave radiometer

data has revealed more problems due to the uncertainties in the ice concentration data itself. Tian-Kunze et al. (2014) has shown that in the regions with high concentrations, correcting the retrieved ice thickness with ice concentration data set with an uncertainty of 5% can cause higher errors than the 100% ice coverage assumption.

The thermodynamic insulation effect of snow is considered in the SMOS retrieval by using a simple statistical relation between ice and snow thickness, i.e. that the snow depth on top of the sea ice is 10% of the sea ice thickness. To estimate the total (snow+ice) sea ice thickness 10% needs to be added to the SMOS sea ice thickness included in the product. The additional snow is neglected in the following for the sake of simplicity except for one example (Table 2).

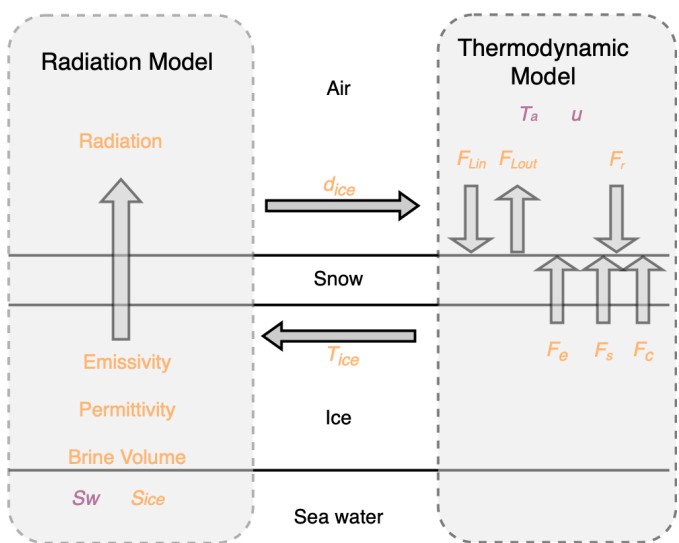

**Figure 3.** Retrieval structure with L-band brightness temperature radiation model and thermodynamic model. Variables in purple are input parameters from auxiliary data, in orange are the calculated parameters within the retrieval model. $F_r$ is incoming shortwave radiation, $F_{lin}$ and $F_{lout}$ are incoming and outgoing longwave radiation, $F_e$, $F_s$, and $F_c$ are latent heat flux, sensitive heat flux, and conductive heat flux respectively.

## 4 Antarctic SMOS sea ice thickness Climatology

SMOS sea ice thickness retrieval is suitable for thin ice detection, i.e. mostly thin first year ice. As ice grows, SMOS brightness temperature gets saturated against ice thickness and the uncertainty increases exponentially with increasing ice thickness. For sea ice thicker than about 1 m, the algorithm only provides an estimate of the maximal retrievable ice thickness (Tian-Kunze et al., 2014). This leads to considerable underestimates of the thickness where the sea ice is thicker than one meter, e.g. in areas of deformed and multi-year ice.

In addition, Antarctic ice drift causes frequent occurrence of leads due to ice divergence. The retrieval assumes 100% ice coverage which leads to an underestimation of the ice thickness in regions with significant fractions of open water like in the marginal ice zone. Nevertheless, the growth and distribution of the seasonal ice around Antarctica is well captured by SMOS.

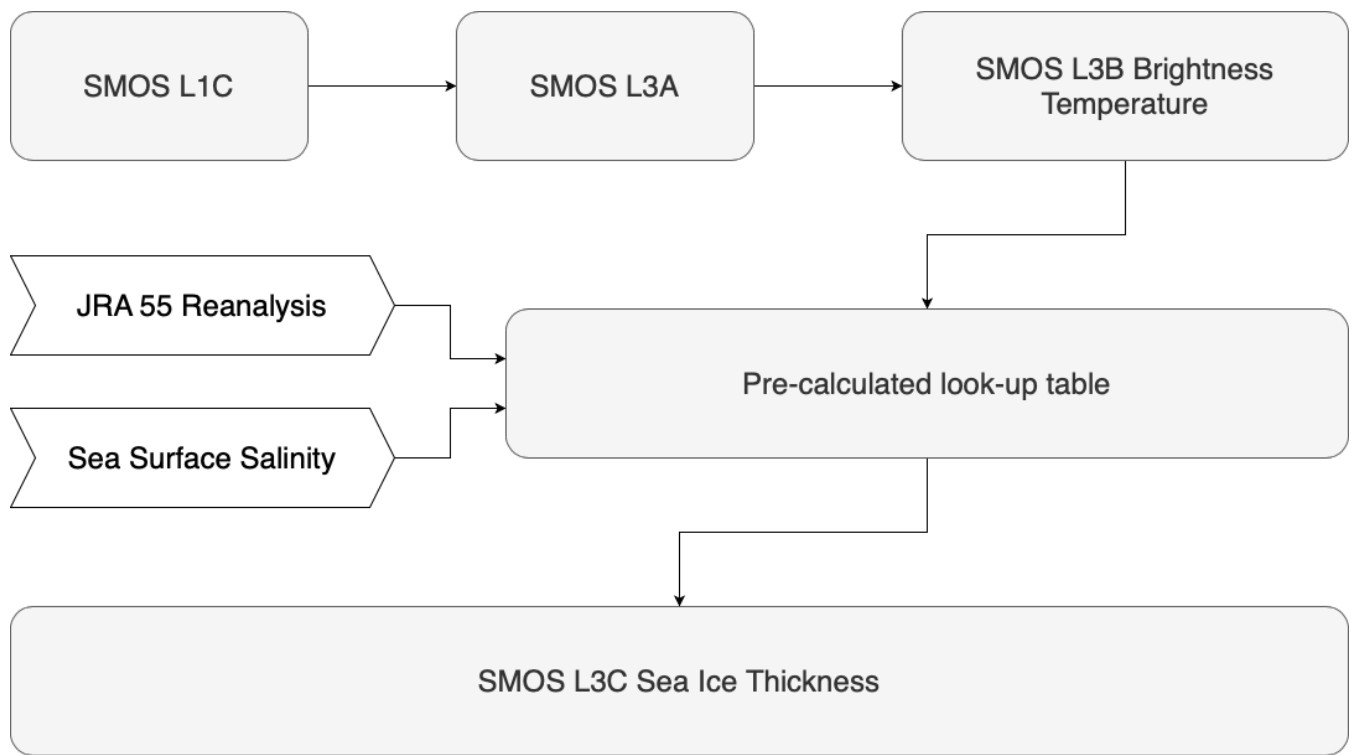

**Figure 4.** Schematic flow chart of SMOS sea ice thickness retrieval.

Figure 5 shows the monthly mean sea ice thickness averaged over the time period of 2010 to 2023. Antarctic ice thickness,
version v3.2, was only retrieved during austral winter periods, from 15 April to 15 October, therefore, for April and October
only a half month is used to calculate the mean. The period was extended for version v3.3 to investigate the range of validity
of the product.

Thick ice is observed throughout the seasons in the western Weddell Sea, where thick multiyear ice dominates. In the freeze-
up period, both ice coverage and ice thickness increases rapidly around the Antarctic coast. Eastern Weddell Sea and Ross Sea
show similar ice thickness conditions from fall to spring, with thick ice more dominant in winter, whereas in fall and spring
considerably large areas of thin ice is present. In Western Pacific and Indian Ocean thin ice is dominant even in winter.

Around Antarctic coast there are frequent polynya openings due to katabatic winds. The SMOS retrieval demonstrates its
advantage detecting polynya areas covered by thin ice (Figure 5).

The anomalies with respect to the total monthly means, the climatology, are shown in Fig. 6 for May to September 2023.

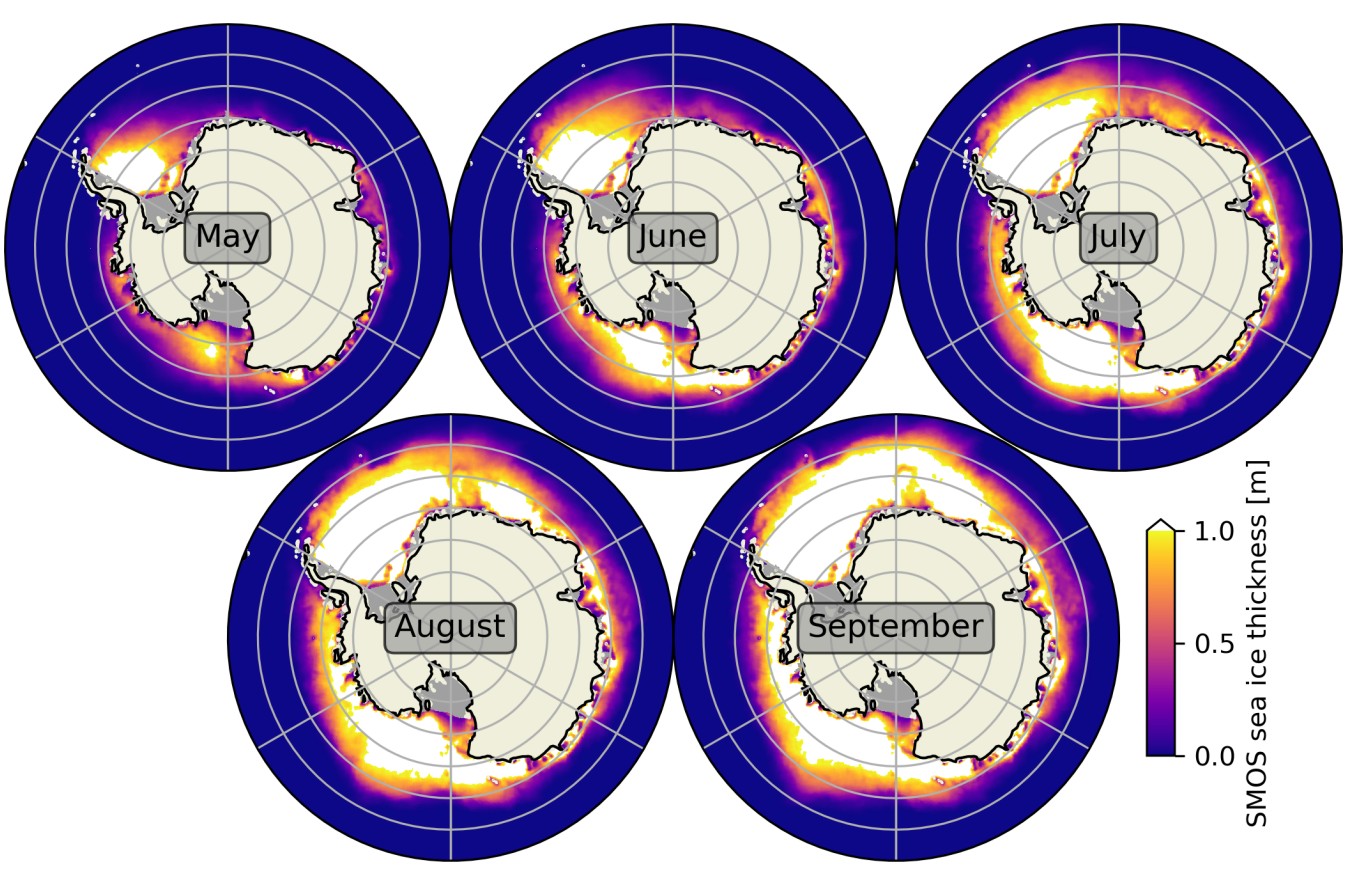

**Figure 5.** Average SMOS sea ice thickness (version 3.3) over the time period of 2010-2023 from May to September.

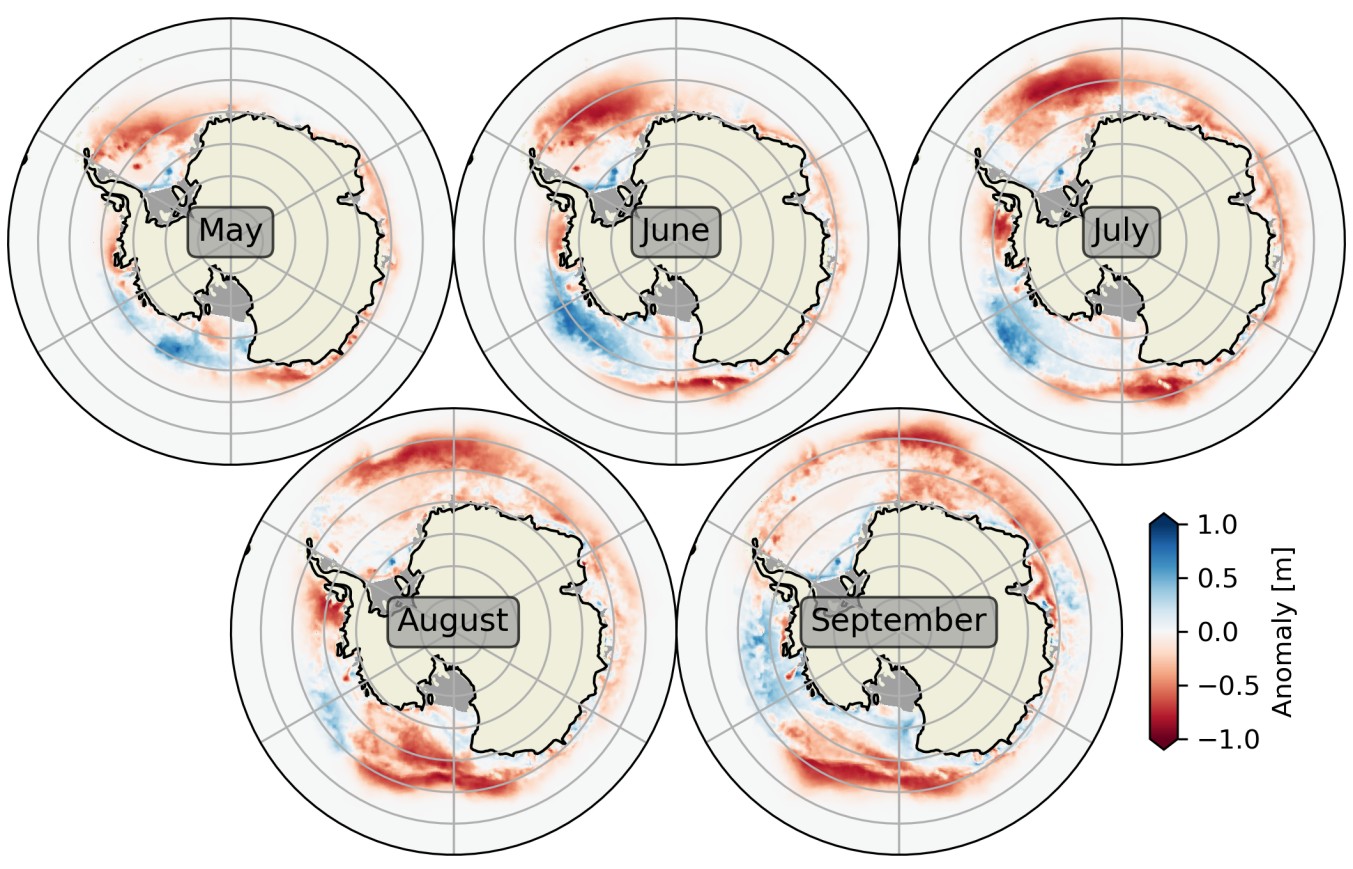

**Figure 6.** SMOS sea ice thickness anomalies from May to September 2023 (monthly means, version 3.3). The reference period is from 2010-2023.

# 5 Validation

## 5.1 Validation with HEM measurements

Sea ice thickness measurements using the HEM-method (Section 2.2.1) cover a spatial scale comparable to the SMOS radiometer resolution and therefore allow direct comparisons for specific days. Four measurement flights have been conducted in areas where the sea ice thickness is within the valid range for the SMOS retrieval. The flight tracks are displayed in Figure 2 and summarized in Tab. 2 together with the SMOS sea ice thickness for the two different versions v3.2 and v3.3. For the comparison, we take two different approaches, on the one hand based on individual SMOS pixels and on the other hand by averaging over the entire flight tracks for the respective days. Although the grid used has a spatial sampling size of 12.5 km, the actual resolution of the SMOS radiometer footprint is significantly lower at about 35 km for nadir and coarser at other angles of incidence. Therefore, the individual SMOS pixels are not independent of each other, which justifies the approach of averaging over the entire flight.

The sea ice thickness pattern from the four HEM flight tracks is well represented in the SMOS data, with the thinnest ice (HEM average 0.52 m) detected on 19 June, 2013 at the southernmost location close to the prime meridian and the thickest ice (HEM average 1.33 m) observed on 7 July 2013 at about 9°W. The correlation coefficient $R$ is 0.7 with a root mean square deviation of 0.26 m between two data sets for $N = 53$ individual SMOS pixels. Both versions v3.2 and v3.3 show similar results but v3.3 exhibits about 5 cm thicker ice and slightly increased RMSD.

Table 2 summarizes the metrics calculated for the averages over the four entire flights as well as for the single pixel. The mean deviation is about 3 - 8 cm for both the pixel-based and flight-based approaches, while the root mean square deviation for the averaged flights reduces to 0.1 - 0.15 m and the correlation coefficient increases. A comparison with a 10% snow layer shows a significant additional deviation and increased RMSD. Based on this comparison to the HEM data we conclude, that the SMOS sea ice thickness should rather be interpreted as the total (snow+ice) thickness.

## 5.2 Validation with SUIT measurements

With an average trawl distance of about 2 km the SUIT measurements sample only a small part of the SMOS measurement footprint and are therefore not representative for inhomogeneous ice conditions. Polarstern's cruise track can also lead to sample selection bias. The open water fraction about 10-20% derived from the SUIT zero thickness values is consistent with an overall reduced ice concentration in the marginal ice zone (not shown). This relatively large proportion of open water can explain the underestimation of the SMOS sea ice thickness based on the assumption of a 100% ice cover. The underestimation compared to the validation data is about 25 cm for the median of the SUIT tracks and 45 cm for the mean (Table 2). The differences between the SMOS sea ice thickness versions v3.2 and v3.3 are negligible.

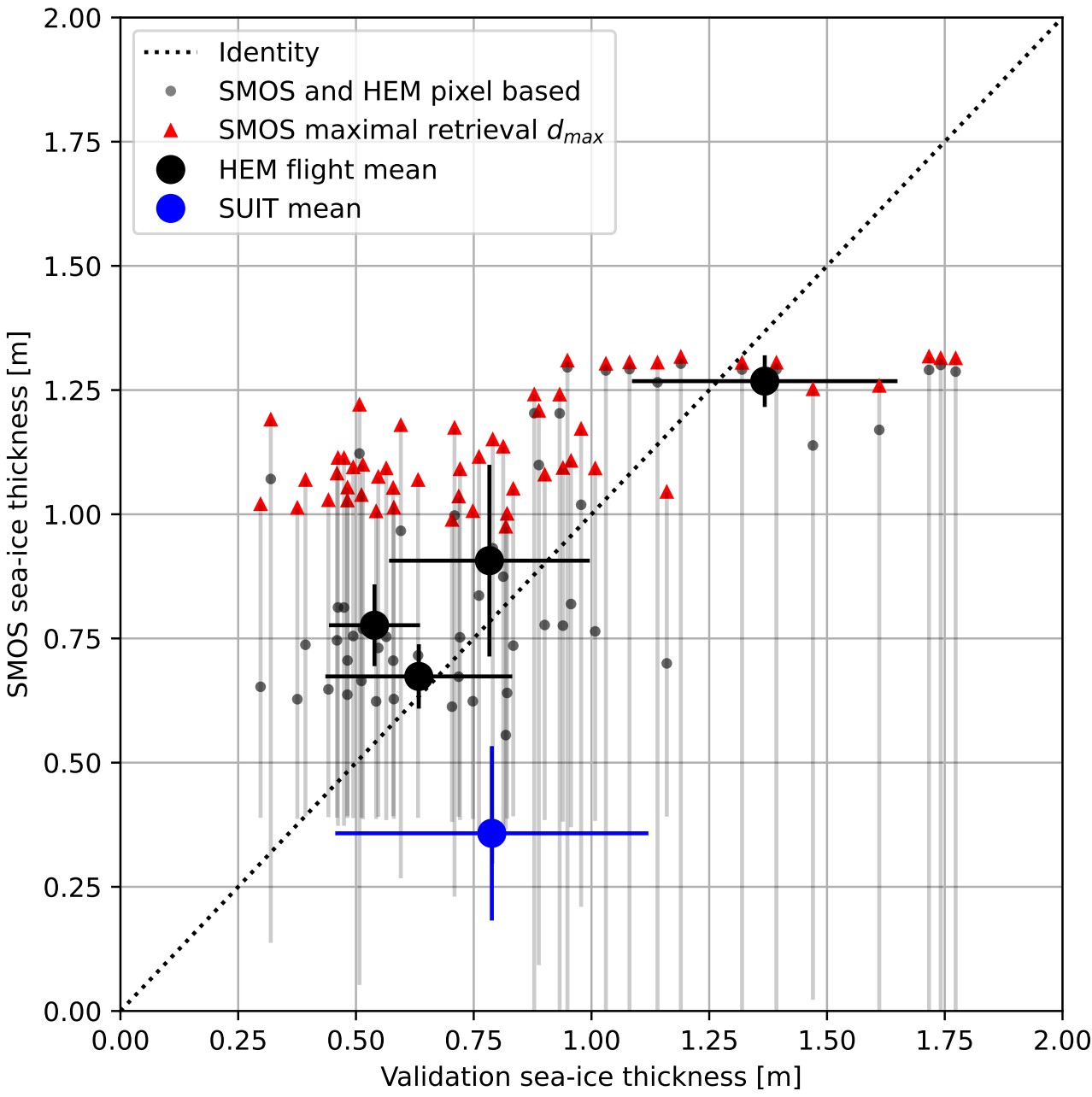

**Figure 7.** Scatter plot of SMOS sea ice thickness and validation data collected in 2013. The graph includes airborne HEM (June/July) and underwater SUIT (Aug-October) thickness data for comparison. The maximum retrievable SMOS thickness is indicated with red triangles for pixel based data points (HEM only). The SMOS uncertainty as provided in the product is given by the lower y-error-bar. The big dots represent the averages over the entire flights and the error-bars their corresponding standard deviation. For SUIT the average over the entire set of measurements is shown with the standard deviation calculated from the averages of 9 tracks.

**Table 2.** Mean deviation (MD) and RMSD of SMOS and validation ice thicknesses in the Weddell Sea, 2013. All length units in cm. Two kind of validation data are used, HEM and SUIT. $N$ is the number of co-located data points for the pixel-based comparison and for the averaged flights, respectively. $R$ is the value of Pearson's correlation coefficient. The comparison v3.3+10% exemplifies the influence of the otherwise neglected snow.

| Time period | Val. | $\mu \pm \sigma$ | SMOS | $\mu \pm \sigma$ | N | R | MD | RMSD |
|---|---|---|---|---|---|---|---|---|
| June-July | HEM | 83±37 | v3.2 | 85±25 | 53 pixels | 0.71 | 2.8 | 26 |
| June-July | HEM | 83±37 | v3.3 | 90±25 | 53 pixels | 0.69 | 7.4 | 28 |
| June-July | HEM | 83±32 | v3.2 | 86±23 | 4 flights | 0.99 | 2.6 | 10 |
| June-July | HEM | 83±32 | v3.3 | 91±22 | 4 flights | 0.96 | 7.5 | 14 |
| June-July | HEM | 83±32 | v3.3+10% | 100±25 | 4 flights | 0.96 | 17 | 20 |
| Aug-Oct. | SUIT mean | 79±33 | v3.2 | 34±16 | 9 pixels | 0.66 | -45 | 52 |
| Aug-Oct. | SUIT median | 59±24 | v3.2 | 34±16 | 9 pixels | 0.71 | -25 | 30 |
| Aug-Oct. | SUIT mean | 79±33 | v3.3 | 36±18 | 9 pixels | 0.66 | -43 | 50 |
| Aug-Oct. | SUIT median | 59±24 | v3.3 | 36±18 | 9 pixels | 0.69 | -23 | 29 |

## 5.3 Validation with ULS data

We group the ULS data in different sets. There are three ULS (ULS set1, Table 3 ) that meet all selection criteria for a comparison (Sect. 2.2.2). Additionally to set1, set2 includes one slightly shorter data set (AWI230). Set3 includes additionally to set2 another ULS (AWI210) in a region with thicker ice. Table 1 includes also the neglected ULS data for completeness. The neglected buoys measured either for too short periods without temporal overlap with SMOS, e.g. AWI209 from December 31, 1992 to November 11 1993, or captured predominantly too thick ice (AWI207, AWI212, AWI 217, AWI233). An example

from 2010 (ULS206 in Fig. 10) shows what a direct comparison looks like with an ice thickness outside the range that can be detected by SMOS. Without showing more of these thick ice examples, we can say that the SMOS retrieval is not reliable for these cases.

Since a direct comparison is not very meaningful due to the low representativeness of the point samples, we first compare the monthly climatologies. Another limitation is the mismatched time period for averaging which is 1996-2010 for the longest

ULS dataset (AWI229) versus 2010-2020 for SMOS v3.2 and 2010-2023 for SMOS v3.3. If we assume a stationary climate, the monthly mean sea ice thickness values and the interannual variability should still be comparable. In fact, there is reasonable good agreement regarding the seasonal cycle (Figure 8) for the ULS within the valid thickness range. The ULS AWI210 measures too thick ice which is mostly outside the valid range for the SMOS retrieval.

The scatterplot (Fig. 9) and the statistical metrics (Table 3) confirm the general validity. With a slight overestimation of

280 2-3 cm, SMOS is very close to the ULS measurements, also the interannual variability is very similar. The mean squared error can be given as 14 cm. With $N = 21$ monthly mean data points, a high correlation of $R = 0.94$ is achieved for v3.2 and with

$N = 36$ a correlation of $R = 0.95$ for ULS set2 and v3.3. Including the thicker ice ULS AWI210 (set3) shows the increased uncertainty of the retrieval for sea-ice thicker than about one meter.

A direct comparison of SMOS with ULS data for 2010 confirms the general picture. Fig. 10 shows the daily mean ULS sea ice thickness together with values from the respective co-located SMOS pixel. Table 4 provides the corresponding statistics. In terms of mean deviation the best agreement can be seen with the ULS AWI229. However, this data set is also problematic due to larger data gaps in the ULS record. A very good general agreement can be seen regarding the freeze-up of ice and open water, which is also accompanied by relatively high correlation coefficients.

**Table 3.** Mean deviation and RMSD of SMOS and ULS sea ice thickness based on monthly long-term "climatological" mean. All length units in cm. ULS set1 consists of ULS 227, 229 and 231, set2 additionally includes ULS 230, and set3 ULS 210.

| Time period | Val. | $\mu \pm \sigma$ | SMOS | $\mu \pm \sigma$ | N | R | MD | RMSD |
|---|---|---|---|---|---|---|---|---|
| Apr. - Oct. | ULS set1 | 46±32 | v3.2 | 49±38 | 21 | 0.94 | 3 | 14 |
| Feb. - Oct. | ULS set1 | 36±33 | v3.3 | 38±40 | 27 | 0.97 | 2.3 | 12 |
| Feb. - Oct. | ULS set2 | 38±36 | v3.3 | 40±40 | 36 | 0.95 | 0.9 | 13 |
| Feb. - Oct. | ULS set3 | 52±50 | v3.3 | 46±43 | 45 | 0.91 | -5.5 | 22 |

**Table 4.** Mean deviation and RMSD of SMOS and ULS sea ice thickness based on the period of temporal overlap in 2010. All length units in cm.

| Time period | Val. | $\mu \pm \sigma$ | SMOS | $\mu \pm \sigma$ | N | R | MD | RMSD |
|---|---|---|---|---|---|---|---|---|
| Feb. - Oct. | ULS 206 | 249±138 | v3.3 | 34±27 | 213 | 0.34 | -215 | 252 |
| Feb. - Oct. | ULS 208 | 125±70 | v3.3 | 74±46 | 214 | 0.68 | -50 | 72 |
| Feb. - Oct. | ULS 229 | 38±37 | v3.3 | 45±53 | 164 | 0.75 | 6.4 | 36 |
| Feb. - Oct. | ULS 232 | 103±68 | v3.3 | 64±47 | 212 | 0.73 | -40 | 51 |

## 6 Discussion

SMOS sea ice thickness in the Antarctic is still a preliminary product with lots of room for improvement. Although it is here derived using the same retrieval algorithm as in the Arctic, the situation in the Antarctic is more complicated and uncertain. This is due to various influencing factors. First, due to Antarctica's remote location and lack of commercial exploitation, there have been fewer applications for Antarctic sea ice thickness. As a result, there is much less observational, in situ, and validation data. Second, the sea-ice conditions are different than in the Arctic and it is unclear how the retrieval procedure needs to be adjusted, e.g. more snow and flooding. Third, this also affects other remote sensing methods and therefore there is much less data to compare. In this first analysis we only compare the SMOS data based on the resulting sea ice thickness. Further analyses

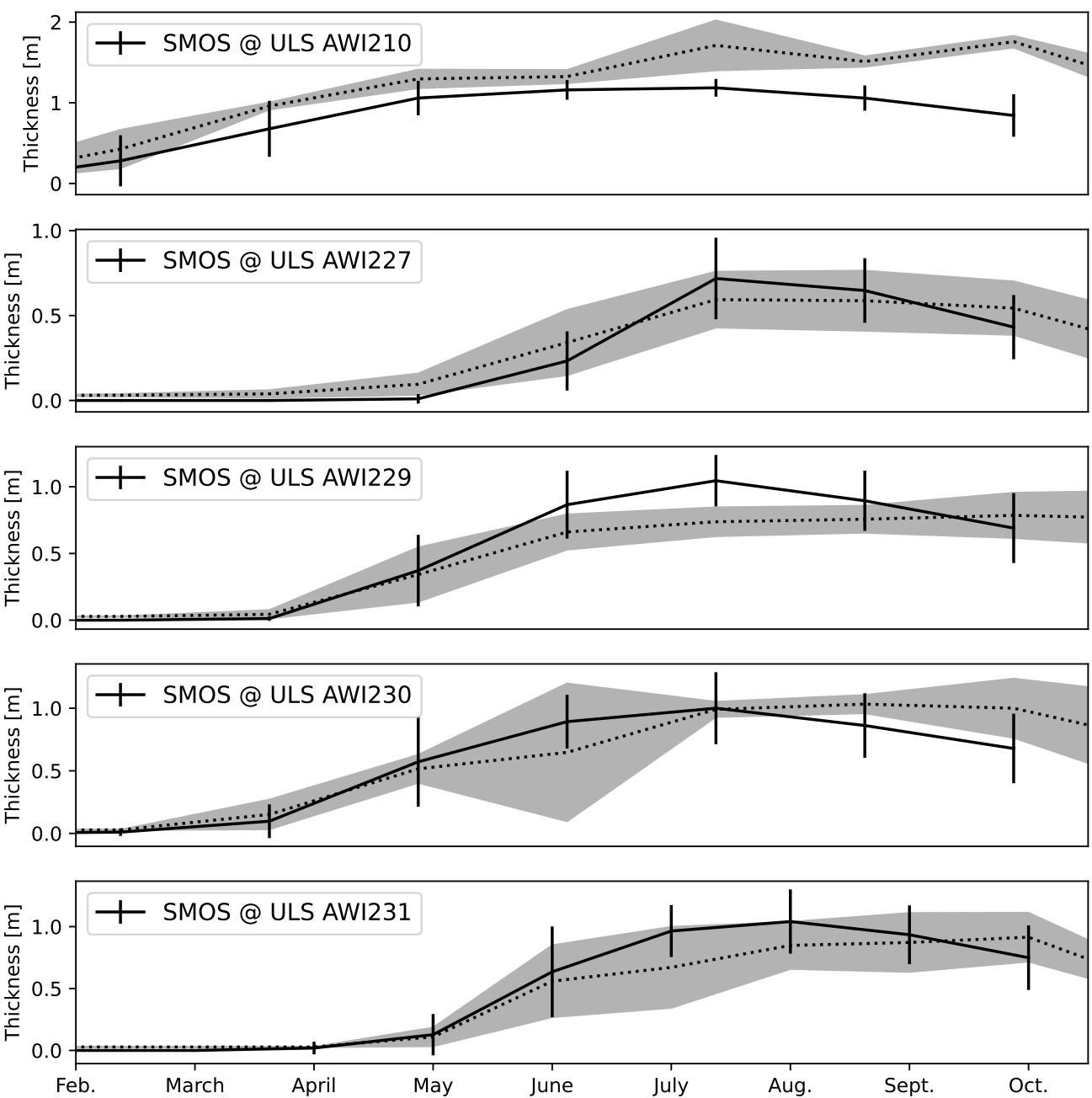

**Figure 8.** Climatological monthly mean ULS and SMOS sea ice thickness version v3.3. The shading and the bars indicate the interannual variability (standard deviation) of ULS and SMOS sea ice thickness respectively.

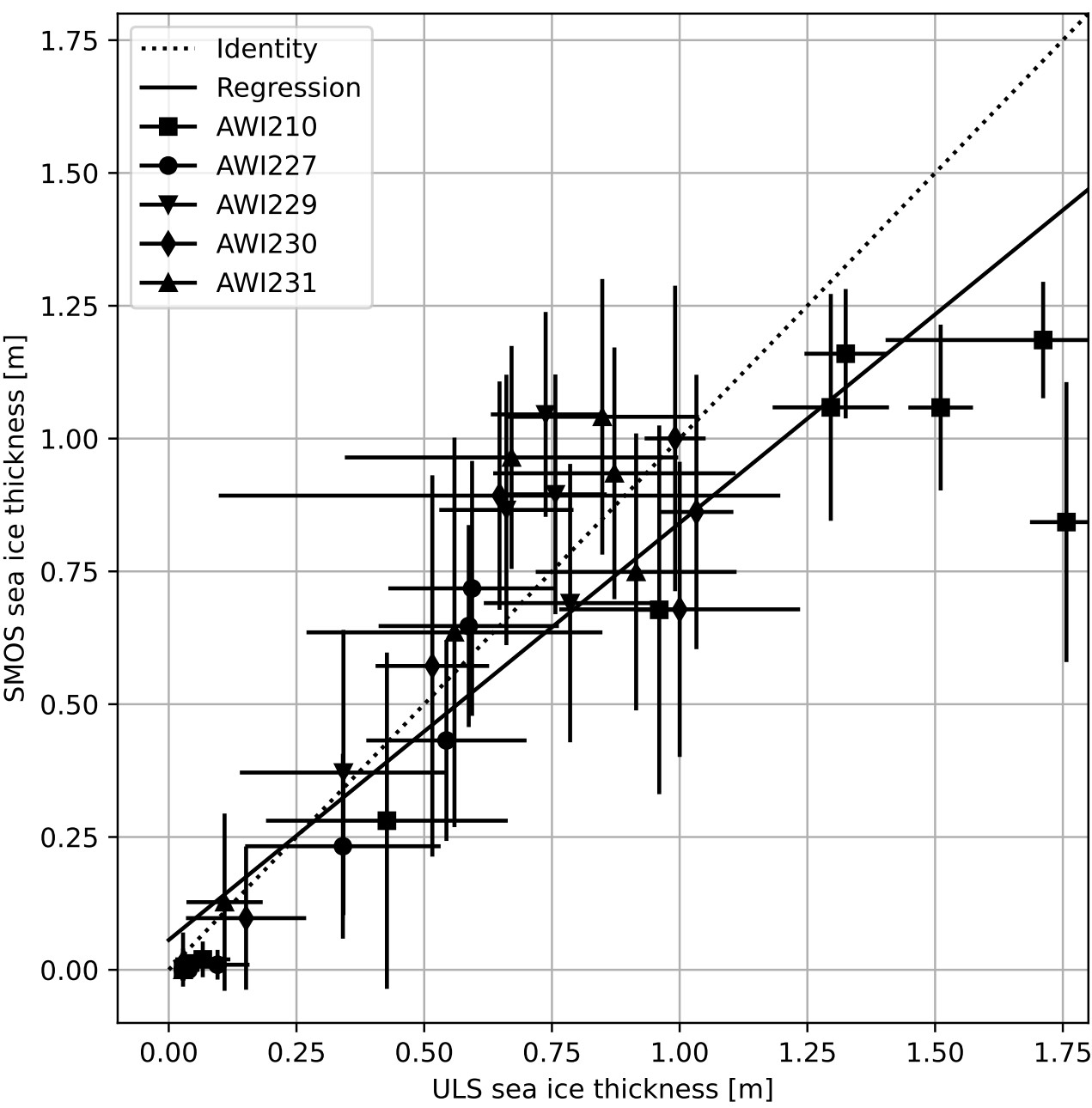

**Figure 9.** Scatterplot comparisons between climatological monthly mean ULS and SMOS sea ice thickness. The bars indicate the interannual variability (standard deviation).

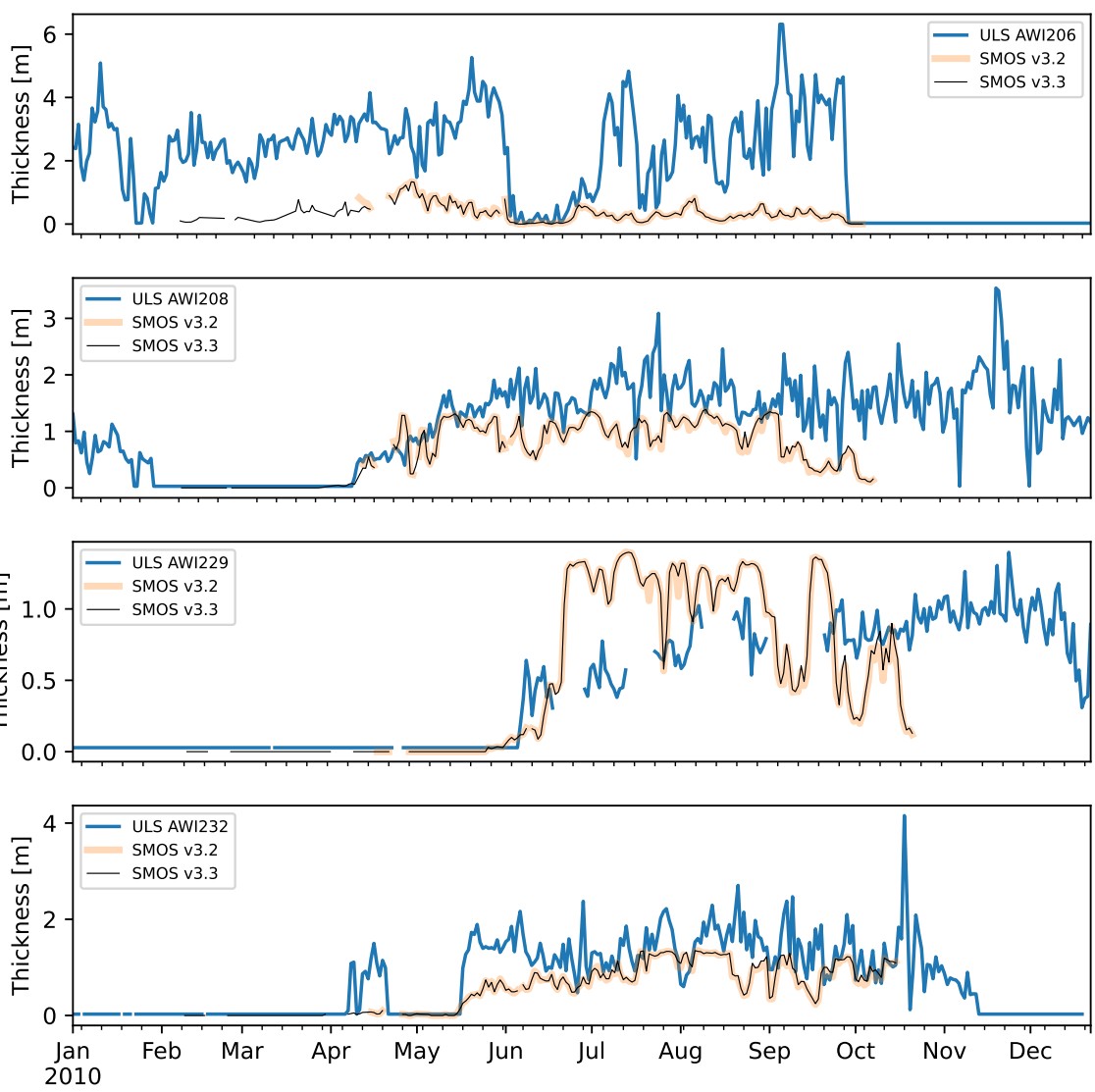

**Figure 10.** ULS and SMOS daily mean sea ice thickness time series for the year 2010. Two different SMOS versions are shown.

regarding the validity of sea-ice physics and retrieval parameterisations have yet to follow. sea ice thickness validation is not universal due to the limitations of SMOS retrieval. The existence of an upper limit for the sea ice thickness retrieval requires a range selection of the validation data with a maximum of about one meter. It is not possible to decide whether the sea ice is thicker than one meter based on the SMOS data alone. The scope of application of the SMOS data is therefore in areas with thin sea ice or in synergetic combination with altimeters. Due to a pre-selection of suitable validation data, only the application for the area of thin sea ice is considered here. An important characteristic of the validation data is its representativity in the SMOS footprint. While the helicopter can measure hundreds of kilometers, the SUIT's use is limited to distances of a few kilometers and thus only captures a small fraction of a SMOS footprint with a diameter of around 35-40 km. It is therefore not surprising that the comparison with the SUIT data showed the worst agreement with the SMOS sea ice thickness. The underestimation at the sea-ice margin is likely to be a real deficit at least in this specific area where relatively thick sea ice is drifting out of the Weddell sea. Little can be done about this problem unless additional sensors are used at the same time to correct for the influence of sea ice concentration. One basic assumption for the sea ice thickness retrieval is a closed ice coverage (100% sea ice concentration) (Tian-Kunze et al., 2014). This assumption often does not hold in the divergent marginal ice zone where the SUIT measurements were conducted. In summary, the SUIT measurements show a substantial underestimation of the SMOS sea ice thickness and the current method is not suitable for capturing this uncertainty. However, since the representativeness of the SUIT measurements is low, these results are not given much significance in the overall assessment.

The HEM flights and ULS measurements are much more representative since they cover larger fractions of the SMOS footprint and longer time periods, respectively. Based on averaged HEM flights and monthly ULS climatologies we find a small mean difference (bias) of less than 3 cm and root-mean-square deviation of about 15 cm with a high correlation coefficient R>0.9 for the valid sea ice thickness range between zero and about one meter. This results gives us good confidence in the overall validity of the SMOS dataset for thin sea ice. In the present version, the SMOS sea ice thickness should be considered as the total (snow+ice) thickness for the best agreement with the validation reference.

## 7  Limitations and issues

The SMOS level 3 sea ice thickness product has some inherent limitations as it opted for a pure SMOS product without the use of additional satellite sensor data. A multi-sensor product is usually referred to as level 4 by definition. A new SMOS level 4 sea ice thickness product for the Antarctic is currently in development and will address some of the limitations of the level 3 product.

### 7.1  General limitations of the SMOS level 3 sea-ice product

A necessary prerequisite for measuring sea ice thickness is the sensitivity of the measured brightness temperature to the thickness. This sensitivity disappears as sea ice approaches its melting temperature and therefore the method is in general limited to the cold seasons. Another general limitation is the coarse resolution of the SMOS measurements, which is particularly

difficult when different surface types are present, for example at coastlines. RFI contamination is another common problem in the Arctic, but is negligible in Antarctica.

## 7.2 Sea ice concentration

The algorithm for the thickness retrieval assumes a sea ice concentration of 100%. The ice thickness is underestimated when this condition is not met (Tian-Kunze et al., 2014). This potential bias is not included in the ice thickness uncertainty which is provided in the product (Tian-Kunze et al., 2014). Future multi-sensor products (level 4) should address this limitation, but the inability to simultaneously measure ice concentration and thickness with SMOS prevents the estimation of this uncertainty.

## 7.3 Snow

Antarctic sea ice generally has a thicker snow cover than Arctic sea ice, which has implications for the importance of flooding and snow-ice formation (Massom et al., 2001). These fundamental differences have not yet been taken into account in our method, which is based on parameterizations developed for the Arctic. Dry and not-saline snow is almost transparent at L-band, therefore, in our retrieval we only consider the insulation effect of a snow layer on the sea ice. While this assumption seems sufficient for the Arctic, it may not be valid in some regions of Antarctica. In the Arctic we assumed a statistical 10% snow depth relative to ice thickness. Assuming a double value for the Antarctic snow cover, the mean relative ice thickness differences compared to bare ice in the ice thickness range of 0-1 m would be about 30%. This does not include the effect of saline snow or snow-ice, which is more difficult to quantify.

## 7.4 Temperature

The emitted brightness temperature depends on the sea ice temperature. A strong simplification of our emission model is that it consists just of one sea ice layer, which does not allow vertical temperature profiles to be taken into account. Another strong simplification is the assumption of an equilibrium with the averaged air temperature over a time period of three days. These simplifications cause a too strong dependency on air temperature which is particular pronounced in thick ice regions for relatively warm temperatures. Therefore, unrealistic decreases in ice thickness can be observed when warm air from the ocean is advected over the sea ice. These are artifacts in the product and not true thickness changes.

## 7.5 Sea ice bulk salinity

The sea ice bulk salinity is described as a function of the underlying SSS (Kovacs, 1996; Tian-Kunze et al., 2014). The use of sea-ice salinity parameterizations needs to be further assessed in general, in particular for the SH. The regional SSS variations in the SH are relatively small compared to the NH with its strong river inflows into the Arctic Ocean and the brackish Baltic Sea. Therefore, we expect that the SMOS retrieval method depends very little on the SSS variability in the SH, but the SSS is on average slightly higher compared to the NH.

## 7.6 Icebergs and shelf ice

One obvious artefact which can be seen in the SMOS climatologies (Figures 2 and 5 ) is the persistent relatively small sea ice thickness at about 75.3°S, 37.3°W in the Weddell Sea in front of the Filchner-Ronne ice shelf. This is the position of the iceberg A-23A (e.g. Paul et al., 2015), which was the largest iceberg in the world until A-76 took over that title in 2021. Icebergs are transparent at L-band frequency due to the very large penetration depth in their almost salt-free ice of meteoric origin (Giovanni et al., 2017). Therefore, huge icebergs correspond to low brightness temperatures and thus cause an incorrect small sea ice thickness in the SMOS retrieval. If these icebergs are mostly stationary like A-23A, then they could be masked out which is not that simple with a static mask if they move. Similar problems exist with the coastal land mask used in the product, which is deprecated in some ice shelf areas. Theses effect has not yet been taken into account in the current SMOS sea ice thickness product version and users should be aware about these artefacts.

## 8 Conclusions

This paper presents the SMOS level 3 sea ice thickness product and its initial validation for the Southern Ocean. The SMOS sea ice thickness version v3.2 is based on the level 1C brightness temperature product version v620 and covers the years 2010 to 2020 during southern winter (defined here as from 15th April to 15th October). The updated product version v3.3 is based on the L1C v724 and contains also the most recent data, including the exceptional 2023 anomaly. The processing period of v3.3 starts two months earlier although the data should be used with caution.

We selected three validation datasets from the Weddell Sea that have different degrees of representativeness: HEM, SUIT, and ULS. Based on averaged HEM flights and monthly ULS climatologies we find a small mean difference (bias) and root-mean-square deviation of about 30 cm with a high correlation coefficient R>0.9 for the valid sea ice thickness range up to about one meter. We conclude that the overall validity of the SMOS sea ice thickness has been demonstrated for thin sea ice up to about 1 m thick. However, compared to measurements outside the valid thickness range, a strong underestimation can be observed.

Certain limitations of the present SMOS sea ice thickness product need to be considered:

1. With SMOS data alone one can not provide an upper-limit estimate of the sea ice thickness. This requires the combination with other sensors such as altimeters. More work is necessary to provide a combined synergy product that covers the full thickness range.

2. The assumption of fully closed sea-ice coverage (100% sea-ice concentration) is often not met, leading to a systematic underestimation of sea ice thickness, especially in areas of ice divergence, such as within the marginal ice zone.

3. The SMOS sea ice thickness v3.3 should be considered as the total sea ice thickness, i.e. snow + ice thickness, even though it is supposed to be the ice thickness only.

4. Icebergs are not flagged and show up as relatively thin sea ice in the product. Likewise, the shelf ice edge is not always up to date.

5. Apart from the auxiliary data, the present SMOS sea ice thickness retrieval algorithm was not adapted for the Antarctic. Thus, there is room for improvement to optimize the method for Antarctic sea-ice conditions.

6. More representative sea ice thickness measurements are needed for validation.

## 9 Code and data availability

The daily Antarctic SMOS sea ice thickness data version 3.2 presented here can be assessed on the PANGAEA open data repository https://doi.org/10.1594/PANGAEA.934732 (Tian-Kunze and Kaleschke, 2021). The currently operational SMOS sea ice thickness data version 3.3 are available through ftp://ftp.awi.de/sea_ice/product/smos/v3.3/sh/ and through the ESA dissemination server https://smos-diss.eo.esa.int/oads/access/collection/L3_SIT_Open. The basis for the SMOS sea ice thickness data is the SMOS L1C brightness temperature product https://doi.org/10.57780/SM1-e20cf57. The ULS data are available from https://doi.org/10.1594/PANGAEA.785565 (Behrendt et al., 2013), SUIT data are available from https://doi.org/10.1594/PANGAEA.902334 (Castellani et al., 2019), HEM data are available from https://doi.org/10.1594/PANGAEA.944879 (Hendricks et al., 2022), JRA55 reanalysis data can be downloaded from https://rda.ucar.edu/data/ (Kobayashi et al., 2015), sea surface salinity data can be obtained from https://www.cen.uni-hamburg.de/icdc (Köhl and Stammer, 2008). The NSIDC regional sea ice index is available at https://nsidc.org/data/g02135. The MODIS thin-ice thickness is available at https://doi.org/10.1594/PANGAEA.848612.

Python/Jupyter notebooks to reproduce all figures and statistics are available on the AWI Gitlab repository: https://gitlab.awi.de/public_repository/smos_derived_antarctic_thin_seaice_thickness and https://doi.org/10.5281/zenodo.11213674

*Author contributions.* Lars Kaleschke and Xiangshan Tian-Kunze wrote the manuscript and analyzed SMOS and validation data. Stefan Hendricks provided HEM ice thickness data. Stefan Hendricks and Robert Ricker contributed to the discussion and provided valuable editorial comments.

*Competing interests.* None

*Acknowledgements.* The work was carried out within the projects "SMOS sea ice processing and dissemination service" and "SMOS Expert Support Laboratory", ESA contracts ARG-003-081/SMOS-2020/AWI and 4000124731/18/I-EF. XT acknowledges Max Planck Institute for Meteorology in Hamburg for providing guest status during the work. The authors would like to thank the anonymous reviewers and the editor for their valuable and helpful comments.

## Appendix A: SMOS sea ice thickness product versions

The differences of the here presented SMOS sea ice thickness product versions are summarized in Table A1. The minor change from v3.2 to v3.3 was introduced mainly due to a new version of the L1C brightness temperatures, while the algorithm for determining sea ice thickness has not changed. Furthermore, the data set v3.3 and its technical documentation is referenced with a DOI issued by ESA (European Space Agency, 2023).

**Table A1.** SMOS L3 sea ice thickness data version differences overview.

|  | v3.2 | v3.3 |
|---|---|---|
| SMOS L1C data version | v620 | v724 |
| Period of data availability | 2010 - 2020 | 2010 - present |
|  | 15.Apr.-15.Oct. | 15.Apr.-15.Oct. |
| Projection | EPSG 3412 | EPSG 3976 |
| Data format | NetCDF v3 | NetCDF v4 |
| Citation | Tian-Kunze and Kaleschke (2021) | European Space Agency (2023) |

## Appendix B: SMOS regional sea-ice extent compared with independent passive microwave data

This section discusses how to check the general quality and completeness of the SMOS sea ice thickness product. For this purpose, a regional extent parameter derived from SMOS is compared with independent standard sea-ice extent products, hereinafter referred to as reference. In the following we use the NSIDC sea-ice index derived from the Special Sensor Microwave Imager/Sounder (SSMIS) and a regional mask from longitudinal boundaries (Meier et al., 2007, 2022). The SMOS sea-ice (thickness) extent is defined as the area covered by all pixels above a threshold thickness of $h_0 = 3\,\text{cm}$ (Kaleschke and Tian-Kunze, 2021). By assuming the reference product is of high quality and consistency, the difference between the sea-ice extent values can measure the quality and consistency of the SMOS sea-ice product. The advantage of the method is the similar spatial resolution and temporal sampling of both independent sensors which allows to identify significant deviations from a mean quality in terms of completeness and consistency.

The plot in Figure B1 allows the quality control in a visual way. The overall consistency is very good and confirms the high quality of the SMOS sea-ice data. Significant deviations can be seen only very seldom with the exception of the first months due to the SMOS in-orbit commissioning phase in early 2010 (Martín-Neira et al., 2016). The negative deviations, viewed as downward spikes relative to the reference, are almost always related to missing data shown on the same chart. same plot. Missing data occurs mostly over the open ocean and not within sea ice and can for example be caused by RFI from radio sources on vessels, or just result from gaps between the orbital swaths. However, the Antarctic is of course much cleaner

with respect to RFI sources compared to the populated Northern Hemisphere. Furthermore, we do see an improvement from version v3.2 to v3.3 which is related to changes in the underlying SMOS L1C brightness temperature data v724 compared to the previous operational version v620. All deviations are negative, in line with the reason of missing data, with one exception for May 27, 2017 and version v3.2. This positive deviation, visible as an upward spike relative to the reference, is caused by corrupted data, a processing artefact, which was unfortunately not detected before the submission of the data to the Pangaea

archive. The new version v3.3 is not affected.

    The final graph Fig B2 shows the total sea-ice extent for the beginning of month from 2010 to 2023. A very good general agreement between the reference and SMOS extent can be seen. Furthermore, in both time series, the decrease and record lows since the year 2021 are noteworthy.

    In summary, the quality control confirms the good quality of the SMOS sea ice data relative to the reference. The first few

445    months of 2010 should be treated with care because of effects from the satellite commissioning phase. In general, rare data gaps could easily be filled with interpolated data if necessary. SMOS sea ice thickness data version v3.3 supersedes v3.2 with better quality in terms of completeness but otherwise with no notable differences.

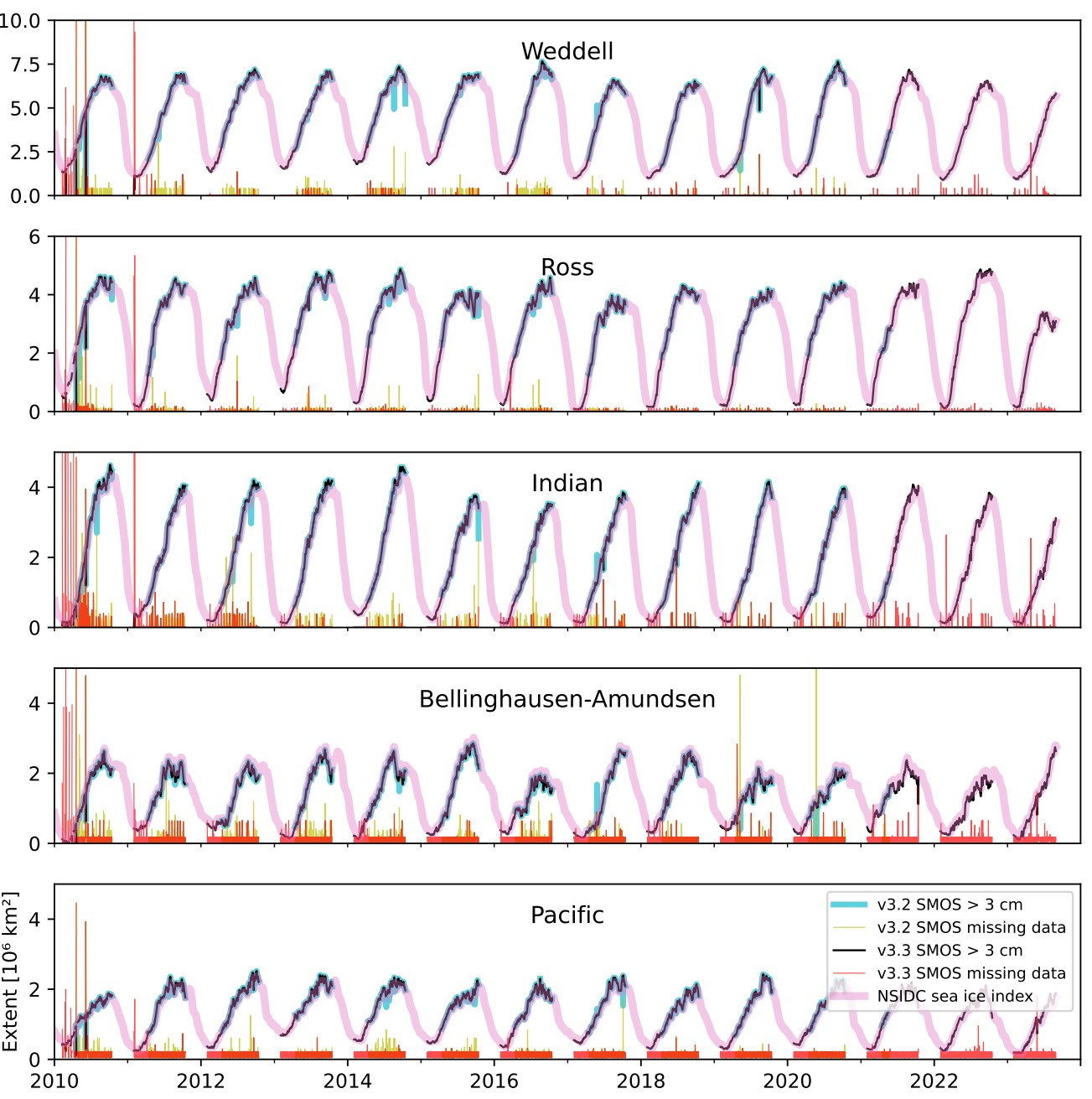

**Figure B1.** Regional sea-ice extent time series used for quality control. The regional NSIDC Sea Ice Index is shown in pink overlaid with the SMOS sea-ice extent in cyan and black for version v3.2 and v3.3, respectively. The yellow and red vertical lines indicate the number of missing data in terms of equivalent area for version v3.2 and v3.3, respectively.

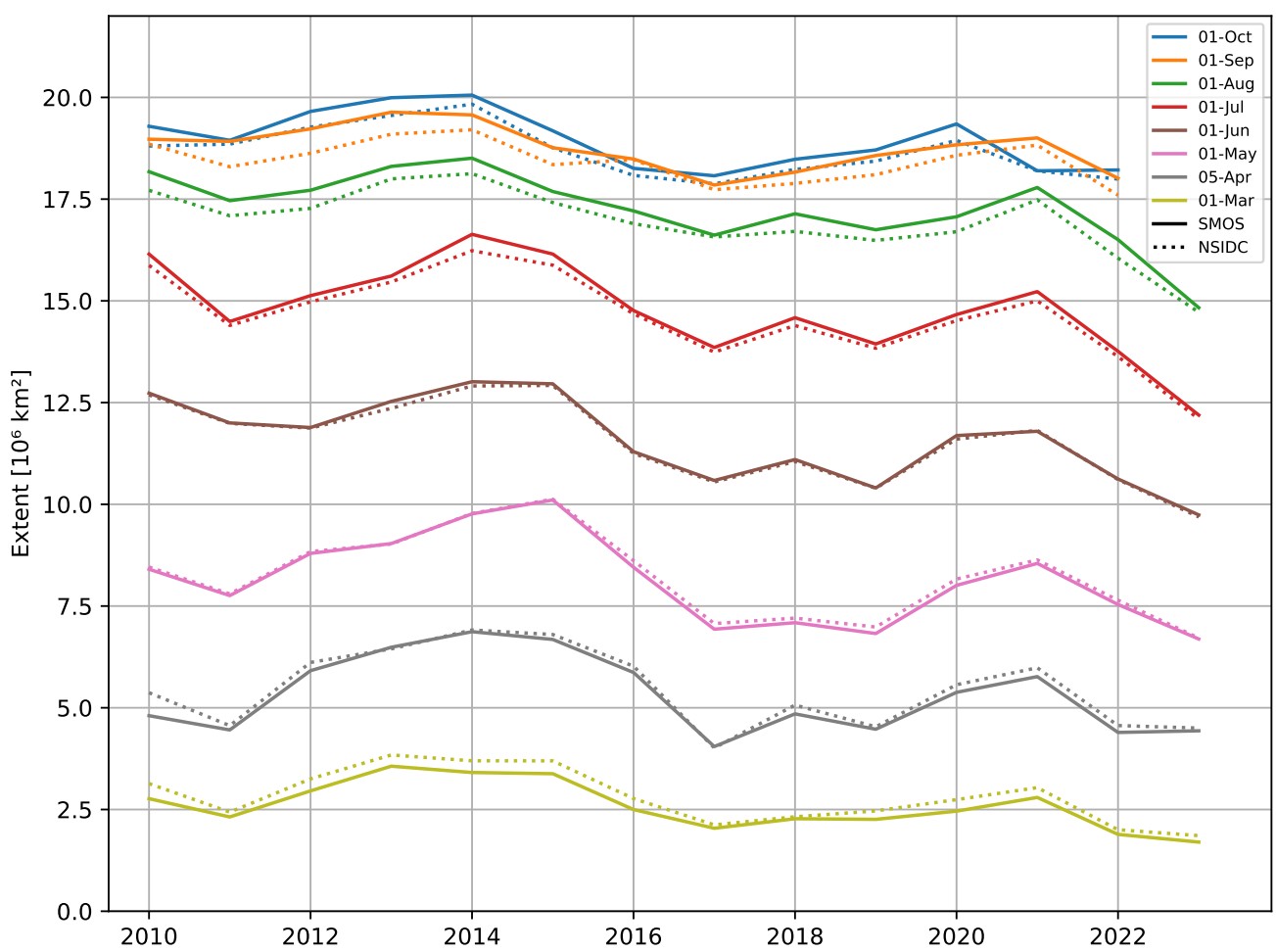

**Figure B2.** Total sea-ice extent time series used for quality control. The NSIDC Sea Ice Index (dash-dotted line) is shown with the SMOS sea-ice extent (solid line) color-coded for different dates, respectively. The extents for the first day of the month are shown. For April the 5th is selected due to missing data for the 1st-4th in 2010. Data for September and October 2023 are not yet available (August 2023).

## Appendix C: Comparison of SMOS and ASPeCt ship-based sea ice thickness observations

For a seasonal comparison, we divide the Southern Ocean into six sectors according to Worby et al. (2008). The seasons summer
(DJF), autumn (MAM), winter (JJA) and spring (SON) are not fully covered by the SMOS product because the method fails in summer and is limited in autumn and spring (Table C1). The ASPeCt data covers the period 1981-2005 (Worby et al., 2008), while the SMOS data covers the period 2010-2023. In addition to the differences in temporal coverage, there are also large differences in spatial sampling: The ASPeCt observations are naturally tied to the ship tracks, while for SMOS the entire sector within the ice boundary is averaged. We assume an ice thickness limit of 0.03 m as the threshold for the sea-ice cover similar
to the usual 15% ice concentration definition of the sea-ice extent (Sect. B).

Given the limitations of the comparison mentioned above, we can state that the two data sets generally agree within the known limits of the SMOS product. The Western Weddell Sea shows the thickest ice followed by the Ross Sea in Winter. SMOS shows a general increase of ice thickness from Autumn to Winter, while this is not the case in the ASPeCt data for the Ross Sea and the West Pacific. We do not wish to interpret these numbers further and refer to the fact that the standard
deviation of the total ASPeCt ice thickness $0.87 \pm 0.91$ m is greater than the mean (Worby et al., 2008).

**Table C1.** Comparison of ASPeCt and SMOS mean sea ice thickness (unit: m) in different sectors. Autumn and spring in the SMOS data are referred to May and September, respectively.

| Seasons | Autumn | | Winter | | Spring | |
|---|---|---|---|---|---|---|
| Sectors | ASPeCt | SMOS | ASPeCt | SMOS | ASPeCt | SMOS |
| Ross Sea | 0.82 | 0.66 | 0.72 | 0.76 | 0.67 | 0.72 |
| Bellinghausen-Amundsen Sea | | 0.36 | 0.65 | 0.51 | 0.79 | 0.52 |
| W. Weddell Sea | 1.38 | 0.88 | | 0.91 | 1.33 | 0.84 |
| E. Weddell Sea | 0.44 | 0.54 | 0.54 | 0.73 | 0.89 | 0.74 |
| Indian Ocean | 0.45 | 0.36 | 0.59 | 0.46 | 0.78 | 0.54 |
| West Pacific | 0.75 | 0.46 | 0.72 | 0.48 | 0.68 | 0.43 |

## Appendix D:  Comparison of SMOS and MODIS sea ice thickness - a polynya case study in the southern Weddell Sea

We compare the sea ice thickness derived from MODIS thermal-infrared imagery with SMOS (Paul et al., 2015). Clouds limit the applicability of the MODIS sensor in particular during polar nighttime (Paul et al., 2015). Furthermore, the MODIS data range was limited to a maximum ice thickness of 0.2m because of substantially higher uncertainties for thicker ice (Paul et al., 465  2015). The average spatial resolution of the MODIS grid is 2 km × 2 km.

Initially, we assessed the MODIS data set by evaluating the number of valid pixels (not masked), aiming to identify days with optimal data availability. This analysis led to the selection of three dates for a case study: April 23, May 6, and May 8, 2012, identified as exemplifying favorable conditions. The masks were not included in the product, so we cannot explain the reasons for the masking, e.g. due to exceeding the thickness range of 0.2m or due to clouds.

The longitude 39°W has been chosen for a profile because it provides a good spatial coverage through the polynya off the Filchner Ice Shelf. The results are shown in Figures D1, D2, and Table D1. It can be summarized that the pattern of the large coastal polynya off the Ronne and Filchner Ice Shelf is captured well in both products. The profile shows an increase of sea ice thickness towards north as expected in such polynya. SMOS overestimates the thickness by about 7 to 9 cm with respect to this specific MODIS product, which is outside the range of the estimated uncertainty of ±4.7cm of the thin-ice thickness 475  derived from MODIS (Paul et al., 2015).

This case is only intended to demonstrate the application of SMOS sea ice thickness data as an example for a polynya, but further discussion and method comparisons would go beyond the scope.

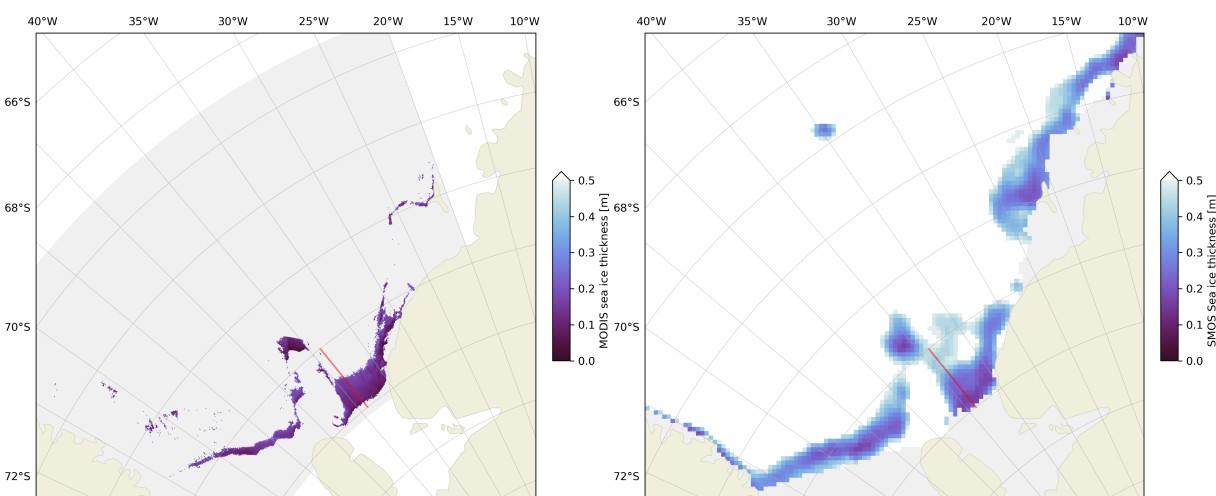

**Figure D1.** Example sea ice thickness derived from MODIS (left) and SMOS (right) on May 6, 2012. The red line indicates the extracted profile as shown in Fig. D2. The grey color indicates the grid of the MODIS data product (left) and the ice shelf mask of the SMOS product (right), respectively.

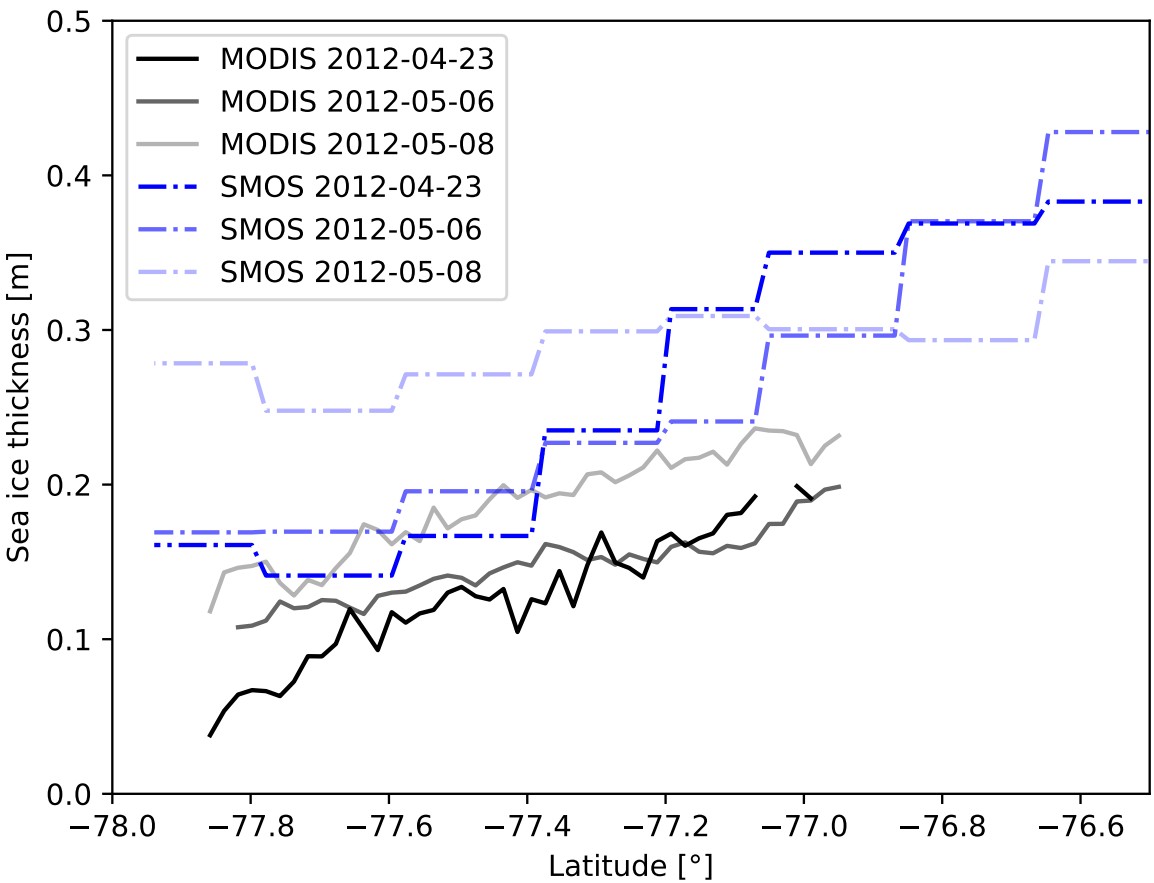

**Figure D2.** Sea ice thickness profiles through the polynya off the Filchner Ice Shelf at longitude 39°W on April 23 and May 6 and 8, 2012.

**Table D1.** Mean deviation and RMSD of SMOS and MODIS sea ice thickness based on the selected profile at longitude 39°W shown in Figures D1, D2. All length units in cm.

| Date | Val. | $\mu \pm \sigma$ | SMOS | $\mu \pm \sigma$ | N | R | MD | RMSD |
|------|------|------------------|------|------------------|---|---|-----|------|
| 2012-04-23 | MODIS | 13±4 | v3.3 | 21±7 | 43 | 0.81 | 8.6 | 10 |
| 2012-05-06 | MODIS | 15±2 | v3.3 | 22±4 | 44 | 0.95 | 6.9 | 7 |
| 2012-05-08 | MODIS | 19±3 | v3.3 | 28±2 | 46 | 0.82 | 9.4 | 10 |

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
