# Peer review of "SMOS-derived Antarctic thickness of thin sea ice: data description and validation in the Weddell Sea"

_Earth System Science Data, 2023_

## Author Response (AR1)

We thank the two reviewers and the editor for their valuable comments and suggestions.

Enclosed you will find our answers, introduced by a general comment.

The text below includes the answers in the online discussion in blue text color (date 15 January 2024) as well as additions for the revised version in dark green. Reviewers' comments are colored in black.

**General answer**

A first general comment seems necessary to clarify the aim of the manuscript. We think that the scope of the present work falls into those of the ESSD journal and like to explain why we selected this journal by first discussing the manuscript and journal aims & scope.

Quote from https://www.earth-system-science-data.net/about/aims_and_scope.html

*Articles in the data section may pertain to the planning, instrumentation, and execution of experiments or collection of data. Any interpretation of data is outside the scope of regular articles. Articles on methods describe nontrivial statistical and other methods employed (e.g. to filter, normalize, or convert raw data to primary published data) as well as nontrivial instrumentation or operational methods. Any comparison to other methods is beyond the scope of regular articles.*

We understand the above in such a way that the introduction of new, improved methods should not be the topic of regular ESSD articles. We described only data obtained using an already published methodology which is in operational use with already existing applications, e.g. https://www.nature.com/articles/s43247-023-00712-w,

We are also working on the algorithm evolution to improve the SMOS retrieval methodology in general and in particular for Antarctic sea ice. However, this ongoing work needs interpretation of data and is thus outside the scope of the journal. In order to justify the use of new, improved methods a comparison to other methods would be necessary and this also would be outside the scope of ESSD as we understood it.

The aim of the manuscript is to describe the validation of the existing SMOS Level 3 sea ice thickness data with all available and suitable ground data. We only consider the more or less direct measurement of sea ice thickness to be a suitable validation measurement. Other satellite measurements are excluded from this. For example, a

comparison with the derived sea ice thickness from the snow or ice freeboard measured with altimeters requires, firstly, strong assumptions about the snow load and scattering horizon (snow backscatter induced freeboard bias) for radar altimeters, secondly, a discussion of the different methods and interpretation of the results. That would of course be a very interesting study and definitely worthwhile to publish, but not suitable for ESSD. There are also other thin ice and/or polynya products from higher frequency passive microwave (SSMI or AMSR) or thermal infrared sensors (MODIS or AVHRR). However, we would consider this a comparison of methods that requires an extensive discussion well outside the scope of ESSD.

Anyway, we did a comparison with MODIS thermal infrared thin ice thickness. This illustrates the application for the Ronne-Filchner polynya, but cannot be considered as validation. For this reason, we have included this in an appendix that can be treated as optional content.

Another example of inappropriate ground validation measurements are sea ice mass balance buoys, as these are typically installed on thicker ice that is already outside the thickness range that can be estimated using SMOS brightness temperatures and have little representativeness for satellite sensor footprints.

We are aware of the significant limitations of the method and since the adequacy and uncertainty of the present methodology are criticized for good reason, we will include an even larger section on "Limitations and problems" in a potential revision. This section then will also include a more extensive discussion of the uncertainties and other parameters provided with the product. The reviewers provided very useful suggestions that will be taken into account.

Finally, we are pleased to announce that a DOI has now also been assigned for the new data that will be updated operationally.

**European Space Agency, 2023, SMOS L3 Sea Ice Thickness, Version 3.3. https://doi.org/10.57780/sm1-5ebe10b**

**Reviewer comments RC1**

Review report of "SMOS-derived Antarctic thin sea-ice thickness: data description and validation in the Weddell Sea" submitted to Earth System Science Data.

The goal of this paper is to construct a data record of Antarctic thin ice thickness from SMOS measurements and the authors try to validate the data record initially. This is an interesting goal and I sincerely thank for the effort of the authors to provide an important ice thickness data set over the Antarctic. However, I have critically wondered that the results of this paper do not fit the scope of the journal which is "the reuse of high-quality data of benefit to Earth system science". The algorithm described in this paper for Antarctic thin ice is just from the existing algorithm applicable to Arctic sea ice. As the authors discussed, the characteristic differences between Arctic and Antarctic sea ice are significant. However, I can't find any effort to consider or analyze the different characteristics between them that can affect the retrieved SMOS thin ice thickness at all throughout the paper. In the case that it will be done in the paper, I can't accept the validation results which are not sufficient for the potential user to use this product having convincing. For instance, thin ice thickness is very important to be used in data assimilation systems in sea ice models. In order to use the ice thickness for this purpose, a much more relevant error analysis should be preceded to provide an observation error covariance matrix. The data produced here may be biased to the real state vectors over different regions. At least, error propagation analysis should be accompanied, however, there is no effort on this in the paper. I totally agree that few observation data compared to Arctic areas. In addition, several assumptions were used to estimate thin ice thickness, however, there is no evidence or error analysis to prove that the assumptions are valid. For instance, the algorithm assumes 100% ice condition which rarely exits over Antarctic thin ice distributions even in the middle of austral wintertime, however, the authors just discussed that less than 100% ice condition doesn't matter because the product shows well growth of the seasonal ice. I can't agree with this. I feel that the authors would compare SMOS sea ice extent to other data in order to show that the SMOS ice thickness data over SMOS is plausible. However, I don't understand why this comparison can provide information on the general quality and completeness of the SMOS sea ice thickness product. As known, sea ice thickness is vertical information while sea ice extent gives horizontal spreads of sea ice. In addition, specific comments are listed below.

Thank you for reviewing our manuscript and for your valuable comments.

We understand your concern. However, we also understand the scope of the journal differently and specifically targeted the manuscript for this journal. The algorithm described in Tian-Kunze et al. (2014) was indeed so far validated only for the northern hemisphere but it is a general method and was not specifically designed for the Arctic. It is true that there are certain limitations and that there are particular uncertainties which we need to describe in more detail. In Tian-Kunze et al. (2014) we conducted an error

analysis with respect to errors in the brightness temperatures, the sea ice salinity and sea ice temperature which is also valid for the present SMOS Antarctic sea ice thickness data. These total uncertainties are provided in the product except the influence of ice concentration as noted in Tian-Kunze et al. (2014). In the present ESSD manuscript we discussed the assumption of 100% ice coverage by confronting the product with the SUIT measurements in the MIZ. In the conclusion we therefore have written *The assumption of fully closed sea-ice coverage (100% sea-ice concentration) is often not met, leading to a systematic underestimation of sea-ice thickness, especially in areas of ice divergence, such as within the marginal ice zone.* However, the improvement of the method is beyond the scope of the manuscript. This requires a different approach using additional sensors, i.e. higher frequency channels for the sea ice concentrations, like those available on AMSR2. Such a new SMOS level 4 synergy product will be of advantage but does not yet exist. This is an area of ongoing research also for the upcoming Copernicus Radiometer CIMR. What we could do to improve the present manuscript is to include an expanded error analysis in a section on 'limitations and issues'.

(1) Page 2, Line 39: 'record' can be 'records'. There are numerous subject-verb agreement problems. I don't want to review all the same problems in this paper. Please check carefully throughout the paper.

Acknowledged, we will correct. Sentence rewritten, and hoping that the excellent Copernicus English copy-editing team will capture potential remaining issues.

(2) Page 2, Line 41: add 'last' between 'since' and 'five' or specify the years you referred to.

Thanks, will be added. Done

(3) Page 2, Line 43: what is the 'can be also be'? If it is a typo, correct this.

Thanks for finding this typo. Removed additional "be"

(4) Page 4, Line 106: does the 'full polarization' mean full Stokes' component including third and fourth Stokes' polarizations? If not, change it into 'first two polarizations'.

Yes, full polarization means SMOS measures the 4 Stokes' components. No change

(5) Page 4, Line 109: This sentence can be modified as "SMOS measures brightness temperatures with a spatial resolution of about 35 km at nadir on a daily basis in the polar regions".

Thanks, that reads better. The suggestion was adopted.

(6) Page 4, Line 116: perhaps, oN and oS can be switched.

Agreed, this matches better. Or we change north and south in the beginning of the sentence. Switched northern and southern.

(7) Page 4, Lines 119-120: Define both 'JRA' and 'GMT' when they first appear.

Acknowledged, will be added. Defined JRA and skipped GMT because it doesn't matter here.

(8) Page 5, Lines 126-127: Again, please define 'GECCO2' and 'MITgcm'

Thanks, will be done. Included definitions of GECCO2 and MITgcm

(9) Page 5, Line 128: 'are' can be 'were'. Please use 'past sentence' when you did something for this work throughout the paper. There are tons of these kinds of issues in the paper. I don't want to spend time to peak all issues.

Acknowledged, we have to go through all of this. Done

(10) Page 5, Line 132: Define 'HEM' and 'SUIT'

It was defined in the Abstract. Maybe we can repeat it. We define abbreviations in the abstract and then again at the first instance in the rest of the text.

(11) Page 6, Line 141: 'ULS' was already defined in the introduction.

Thanks, it was defined three times, we will change. Removed multiple definitions.

(12) Page 6, Lines 141-142: 'have' can be changed into 'had'.

Correct, will be changed. Changed sentence

(13) Page 7, Line 164: how to prescribe snow depth, snow density, and water density in order to utilize the hydrostatic equilibrium equation.

Thanks, we will clarify. We removed the hydrostatic equilibrium from the sentence. Castellani et al. (2019) refer to the total thickness that includes both sea ice and snow.

(14) Page 7, Line 170: the authors mentioned that "in addition, ice thickness variation within the SMOS grid are considered". why did you consider this for what?

Acknowledged, we will better describe this parameterization of thickness variation. We added an explanation for the use of the statistical distribution parameterization.

(15) Page 7-8, Line 176-177: This sentence is imperfect. Please rewrite it. This sentence gives weird information that "inhomogeneities are much smaller than the wavelength of 21 cm."

Thanks, we will rewrite to clarify. Rewritten and simplified.

(16) Page 9, Line 193: Define 'ISEA 4H9'.

Thanks, we will define the ISEA Icosahedral Snyder Equal Area grid and look for a good reference. Defined ISEA

(17) Page 14, Line 233: the authors should discuss the difference between v3.2 and v3.3 in detail in the paper.

Agreed, we will add a detailed description. The version differences v3.2. and v3.3. are now described in an appendix with a table for a clear overview.

(18) Page 15, Lines 261-262: ULS sees different ice floes as sea ice flows by ocean current. In order to make a comparison between ULS and SMOS ice thicknesses, the authors should track the ice floes even for monthly comparison, which process was done in many papers for Arctic sea ice studies. Perhaps the Lagrangian tracking method would be useful.

Understood, but such a tracking is difficult to perform with the long-term historical data and not very promising for the coarse resolution. However, we will further discuss uncertainties related to the sea ice motion.

(19) Regarding the sea surface salinity (SSS) dataset: the SSS climatology used in this study was based on a model simulation over the years 1952-2001 which is far from the SMOS period. Is there any other SSS climatology covering the SMOS observation period? If so, recommend replacing the SSS dataset.

Acknowledged, we will discuss the influence of SSS in the new section "limitations and problems" although it is probably of negligible magnitude. Discussed

(20) Regarding the ULS dataset: the authors converted the ice draft into total thickness using an empirical linear fit equation, which is a very important relationship in the validation section. It is weird that there is no discussion of this fit equation in the paper.

Respectfully disagree, a discussion can be found in Behrendt et al. (2013, 2015) and references therein. We leave this as is..

(21) Table 1: why several ULS data were neglected in this paper? I don

Understood, although the above sentence seems to be broken. We will describe in more detail the selection criteria which is based on the length of time series and the covered thickness range. However, all results and figures can be reproduced using the code provided in the repository. This also includes the selection procedure, i.e. the routines that read and select the ULS data. We added a few sentences:

The neglected buoys measured either for too short periods without temporal overlap with SMOS, e.g. AWI209 from December 31, 1992 to November 11 1993, or captured predominantly too thick ice (AWI207, AWI212, AWI 217, AWI233). An example from 2010 (ULS206 in Fig. 10) shows what a direct comparison looks like with an ice thickness outside the range that can be detected by SMOS. Without showing more of these thick ice examples, we can say that the SMOS retrieval is not reliable for these cases.

Reply

Citation: https://doi.org/10.5194/essd-2023-326-RC1

**Reviewer comments RC2**

This article, titled "SMOS-derived Antarctic thin sea-ice thickness: data description and validation in the Weddell Sea", introduces the sea ice thickness (SIT) product based on L-band radiometer of SMOS for the Weddell Sea. The results presented include promising results and sound validations with various independent observations, and the capability of SMOS for SIT retrieval in Southern Oceans is demonstrated. The scope of the work falls right into those of the ESSD journal. And furthermore, the data link works totally fine, with the data files formatted according to the regulations widely used by the climate science community. However, I do have the following concerns and comments which need further clarification and possibly revisions.

Thank you for reviewing our manuscript and for your valuable comments. We agree that the scope of the work falls right into those of the ESSD journal (see general comment above).

First, I find it worthy a second consideration for the paper's focus on Weddell Sea. The dataset provided covers the whole Antarctic realm. And arguably, Weddell Sea is possibly not the best place for thin ice retrieval, given the dominant thick, perennial ice in the region. I understand that most validations are carried out in Weddell Sea, but what limits the validation in other part of the Southern Ocean? In the Indian/Pacific sector, there are also many available data from ASPECT. And many MIZ campaigns provide SIT over the thin ice, which could be invaluable source of validation especially for SMOS's retrievable range.

This comment addresses a key limitation, the sparsity of suitable validation data. We briefly discussed this already in the introduction and mentioned the ASPeCt program. There are several issues with this ship-based record regarding the representativeness of the visual point observations and potential preferential sampling biases. There are not many campaigns suitable for comparison with the SMOS data because it was launched only in 2010 and the majority of the ASPeCt data has been collected earlier. Moreover, ships usually don't go there during the cold and dark Winter season but prefer the period between November and February/March. However, we could include a further comparison with ASPeCt mean sea ice thickness observations like Giles et al. (2008).

We added a comparison with the ASPeCt data in an appendix.

Second, I consider that the retrieval result needs more in-depth analysis. For example, with some inspection of the daily SIT fields, one can easily discover large temporal (i.e., day-to-day) variability of SIT across the whole field. The (unrealistic) fluctuation of the retrieved SIT is evidently of atmospheric origin, possibly due to passing cyclones and the ensuing thermodynamic signals in the sea ice and snow. The effect of synoptic systems is more pronounced on thick ice (SIT>1m), but potentially present on thin ice as well. This observed phenomenon is directly linked to the thermodynamic quasi-equilibrium assumption (or the lack of) for the retrieval algorithm, as well as the timing of SMOS's passes. I think an immediately available analysis is to explore what is the optimal retrieval interval by SMOS, which is definitely above (coarser than) 1-day. Also, collaterally, how the monthly mean SIT should be computed: mean of SIT or SIT based on mean TB?

We agree that the sensitivity to the auxiliary atmospheric boundary field should be explained in more detail. This is also a key factor for future improvements of the method which we think is beyond the scope of the manuscript.

However, we now recognize that secondary parameters which are included in the provided data set are not described in a way that users can take advantage of them. This includes the "saturation ratio" which is related to the "maximum retrieval sea ice thickness" . These parameters depend on the auxiliary atmospheric field and can be used to identify data of doubtful quality for example when the condition (high saturation ratio) suggests that the real thickness could exceed the maximum retrieval sea ice thickness. The dependency on the atmospheric auxiliary data is much more pronounced in the sea ice thickness range with relatively high uncertainty, i.e. when the thickness approaches the maximum retrieval thickness. This is in fact sometimes obvious in areas of thick sea ice like in the western Weddell Sea when passing cyclones change the temperature. That said, we emphasize that the greatest use of the SMOS sea ice thickness data is for relatively thin ice and not for the thick sea ice which comes with large uncertainties. The dependency on the atmospheric auxiliary data is less pronounced for thin ice which is significantly smaller than the maximum retrieval sea ice thickness.

When calculating the mean SIT, we have to take into account a non-linear relationship and therefore cannot use the mean TB calculated over a month. This relationship, e.g. as shown in Fig. 5 in Tian-Kunze et al. (2014) requires that the time scale for averaging should match the time scale of measurable sea ice thickness with corresponding brightness temperature change in a given resolution cell. We assume that this time scale is approximately one day for SMOS resolution, but would be shorter for higher spatial resolution or strong dynamic conditions.

We conducted an extensive analysis using a number of sea ice mass balance + snow thickness buoys and simulated the brightness temperature using radiative transfer models along their trajectories, including ice concentration from AMSR2. This took us some time and we therefore asked for an extension of the deadline for the revision. We initially thought that this could be included for a discussion of the observed SIT fluctuations but we realized that this would lead to an entire new manuscript, not suitable for ESSD. Therefore, we will now just briefly discuss the existing known problems in the new section limitations and issues. However, the analysis was insightful and will be used for future work on improvement of the method, e.g. for the snow parameterization and determination of the sea ice surface temperature.

Third, a related issue to my second comment is, a better quantification of uncertainty and a clear indication of the usability of the SIT product are needed. For SIT over 1m, the SIT product contains no information. Besides, very large uncertainty is present for SIT over 50cm. Whether the cyclones affects the uncertainty (see above) and whether this

uncertainty is accounted for is in unknown. Furthermore, with a better quantified uncertainty, a simple field of confidence of SIT can be provided, for example: the relative uncertainty lower than 30% (high confidence), lower than 60% (low confidence), or higher than 60% (very low confidence). This would facilitate downstream users who are not experts of the data retrieval. The specific threshold values and the terms could be chosen more carefully, but I do suggest adding such information to avoid potential misuses of the data.

This is a very good comment from a user perspective and one that we will certainly take into account in future developments of the product. As mentioned above, we have learned that a better description of the secondary parameters including uncertainty would be helpful.

Fourth, an outstanding issue of the retrieval method is that it is originally developed for the Arctic. With potential difference in snowfall rate and snow stratigraphy between the two poles, some assumptions may not hold and need justification. At least, the uncertainty caused by them should be estimated. They include: 10% snow depth of sea ice depth, salinity in snow, as well as the potential of snow-ice.

Without large structural changes of the sea ice emissivity model we can not assess the sensitivities to salinity in snow and the potential of snow-ice. This unfortunately has to remain for future work with a new multi-layer model and new sea ice physics parameterizations. Currently we use just one sea ice layer and consider only the thermodynamic effect of snow. What we can do for the present manuscript is to discuss the sensitivity of a changed snow thickness parameterization. In the figure you can see the impact of varying the snow thickness between 10% and 30%. In general a higher snow load leads to a reduction of the retrieved sea ice thickness. The impact of a change from 20% to 30% is much less pronounced compared to the change from 10% to 20%. With this analysis we can now quantify the potential impact of the snow depth assumption for different sea ice thickness categories.

[Figure]

Figure: Sensitivity of sea ice thickness depending on the snow layer parameterised as a fraction (10%, 20% and 30%) of the sea ice thickness. Example calculation for the year 2017 and the entire Antarctic Ocean including all pixels with a saturation ratio < 95%.

We also tried to further investigate the effect of salinity in snow, as well as snow-ice. The findings are however very preliminary, not yet mature enough to be published. In fact, this area of research requires much more basic research. We are still a long way from making clear statements. There is new literature on this topic, but so far it has only been published as a preprint (Discussion started 06 Feb 2024) and the results have not yet been replicated.

Zhou, L., Stroeve, J., Nandan, V., Willatt, R., Xu, S., Zhu, W., Kacimi, S., Arndt, S., and Yang, Z.: Quantifying the Influence of Snow over Sea Ice Morphology on L-Band Microwave Satellite Observations in the Southern Ocean, EGUsphere [preprint], https://doi.org/10.5194/egusphere-2024-81, 2024.

Besides, I find that Section 6 is not an integral part of the paper which focuses on SIT. I suggest moving it to a more proper place, for example, as an appendix.

We agree and will move section 6 to an appendix. Done

There also exist some minor issues, with examples below:

l122: "days temperature" should be "days' temperature" Thanks, will be fixed. Done

l293: missing "." Thanks, will be fixed. Done

Reply

Citation: https://doi.org/10.5194/essd-2023-326-RC2

---

## Author Response (AR2)

We thank the two reviewers and the editor for their effort in re-reviewing the revised manuscript and their valuable comments and suggestions. Below you will find our answers in blue. In summary, we only changed the title and gave answers to the questions. Furthermore, we added a DOI to the Zenodo archive in the section code and data availability.

Thank you to the authors for revising the manuscript according to the guidelines of the two reviewers. There remain some outstanding comments, but these are minor. I summarise them here:

1) The revised title sounds a little unnatural now; the original was perhaps better.

Agreed, we changed the title back.

2) A discussion on the effect of motion on the ULS validation was promised in the response but not found in the revised manuscript.

The effect of ice motion was discussed in Behrendt et al. (2013). We cannot correct this error because ice drift information from satellites is not available with the required high sampling rate. However, this error is certainly small compared to other sources of error, in particular the large discrepancy in the sampling footprints (SMOS 35-40 km vs USL 6-8 m). Furthermore, Behrendt et al. (2013) state that the errors are considered smaller over longer periods of time, such as daily or monthly averages. We would therefore prefer not to discuss this effect further because it is already discussed in detail in the references.

Behrendt et al. (2013): *A problem with the presented ULS data is the lack of ice drift information. Contrary to the measurements reported by Melling et al. (1995), the AWI ULS instruments were deployed without acoustic Doppler current profilers (ADCP). Their data can therefore not be converted into space-referenced data. Ice draft distributions of time-referenced data may contain peaks different from the distributions of space-referenced data (Melling et al., 1995). This sampling problem is induced by the character of the ice drift. If, by chance, only thick ice classes are present when the ice drift is slow, these classes will be more common in the draft statistics. The differences between the two distributions are expected to decrease in daily or even more in monthly mean sea ice drafts. However, the conversion of the observations into regular intervals of space would eliminate the problem.*

3) Line 181: 'the algorithm' can be changed to 'the presented algorithm'

Unfortunately, we can not find this term in Line 181.

4) Chapter 7.4: The authors mentioned that warm air intrusions cause an unrealistic ice thickness estimation. How about the influence of cold air advection? Does the estimated sea ice thickness become overestimated?

We mention only the warm air intrusions because this effect is indeed only in one direction over thick ice because of the non-linearity. For the case of thick ice, which is thicker than the maximum retrieval thickness, warming has an effect on the result because it reduces the inferred maximum retrieval thickness. Cooling (over thick ice) would have no effect on the resulting thickness because the actual thickness is already thicker than the maximum. Over thin ice, however, the effect of ice temperature is much smaller and in both directions. The idea of the algorithm is to correct for ice temperature changes by considering the atmospheric temperature variation in the thermodynamic and emission model. Actually, these variations are taken into account when determining the ice thickness, unfortunately only for the range of validity of the assumptions. The general error associated with uncertain ice temperatures is already discussed in Tian-Kunze et al. (2014). Therefore, we consider the description in Section 7.4. sufficient to describe the current limitation.

---

## Author Response (AR3)

Thanks to the authors for their response - this is sufficient.

However I can still see the "new" title in your revised manuscript - Could you make sure to revert to the original title for your resubmission? The original title was "SMOS-derived Antarctic thin sea-ice thickness: data description and validation in the Weddell Sea"

Thank you and sorry for the confusion. I now really changed the title back.